# Optimal Base-Stock Inventory-Management Policies of Cement Retailers under Supply-Side Disruptions

Manik Debnath [1], Sanat Kr. Mazumder [1], Md Billal Hossain [2,3], Arindam Garai [4,*]
and Csaba Balint Illes [5]

1  Department of Mathematics, Indian Institute of Engineering Science and Technology, Shibpur,
   Kolkata 711103, India; 2021map003.manik@students.iiests.ac.in (M.D.); sanat@math.iiests.ac.in (S.K.M.)
2  Doctoral School of Economic and Regional Sciences, Hungarian University of Agriculture and Life Sciences,
   2100 Godollo, Hungary; shohan_bd13@yahoo.com
3  Business Management and Marketing Department, School of Business and Economics,
   Westminster International University in Tashkent (WIUT), Tashkent 100047, Uzbekistan
4  Department of Mathematics, Sonarpur Mahavidyalaya, Rajpur, Kolkata 700149, India
5  Hungarian National Bank—Research Center, John Von Neumann University, 6000 Kecskemét, Hungary;
   illes.b.csaba@uni-neumann.hu
*  Correspondence: fuzzy_arindam@yahoo.com; Tel.: +91-9932-890-115

**Abstract:** The current study aims to identify some optimal base-stock inventory-management policies that maximize the expected long-run profitability of cement retailers under potential supply-side disruptions. Unlike existing articles, the proposed economic order-quantity model considers periodically varying random demand rates of deteriorating items together with partially back-ordered shortages in the face of those random disruptions. This study computes the global concavity to execute the exemplary aspect for the optimal base-stock level under a slew of cost components and a fixed cycle length. Regarding the optimal pricing-related policies, this study proposes that cement retailers should stock from nearby supplier points. Unlike existing articles, we find that changes to either the unit-holding cost or the unit-lost sales cost have hardly any determining effect on the long-run profitability of retailers. When supply-side disruptions are more likely to occur during peak seasons, this study advocates for a planned capacity addition and higher base-stock levels of cement bags.

**Keywords:** periodic-review inventory model; base-stock policy; supply-side disruption; varying demand rate; deterioration; partial back-ordering

**MSC:** 90B05; 90B06

## 1. Introduction

A number of recent events, including the COVID-19 outbreak in 2019, the Second Nagorno–Karabakh War in 2020, and the ongoing Russia–Ukraine conflict in 2022 have caused massive supply-side disruptions in the Asian thermal coal market. Due to the persistently strong demand and limited supply amidst the ongoing coal crises, the Newcastle coal benchmark index surged over 150% compared to the previous year, surpassing the USD 400 per tonne threshold in December 2022. Likewise, the price of the spot physical coal at Newcastle Port in Australia was USD 436.71 per ton, an all-time high in September 2022 [1]. The crises, as mentioned earlier, can potentially jeopardize the cement industry, which is already engulfed in crisis, thus affecting the price and availability of cement in all local markets across the globe. Industry experts believe cement is a major component in infrastructural growth, like roads, bridges, dams, buildings, canals, and houses. Interestingly, the cement market globally is expected to grow at a CAGR of 5.1% between 2022 and 2029, from USD 340.61 billion in 2022 to USD 481.73 billion by 2029 [2]. Moreover, the

consumption of cement indicates how any nation marks its progress. Thus, any potential disruptions in the cement industry are troublesome for the entire economy of a nation.

Comparable to the current cement crisis, modern industrial history reports various other instances of the devastating impacts of supply-side disruptions that are caused by a lack of robustness and resiliency in existing organizational strategies [3]. For example, Samsung, in 2017, had to stop the production lines of the highly popular Galaxy Note 7 because of the defectiveness of the batteries, which were delivered by its supplier [4]. At that time, Hurricane Harvey unleashed its fury upon the Gulf Coast in August 2017, upending the lives of over 13 million individuals across Texas, Louisiana, Mississippi, Tennessee, and Kentucky. The aftermath of this catastrophe resulted in an astounding economic loss of approximately USD 180 billion. Similar events have reverberated across the globe, underscoring the vulnerability of our interconnected world. Think of Hurricane Maria's devastating onslaught in 2017, the seismic upheaval, and tsunamis that rocked Indonesia in 2018, and more [5]. These incidents not only exact a toll on lives and livelihoods but also spark an urgent demand for recovery efforts. One crucial resource that emerges as a linchpin in the recovery process is cement. This versatile building material is essential in rebuilding shattered communities and infrastructure. However, these dire circumstances frequently trigger disruptions in the supply chain. Earthquakes, floods, transportation hiccups, and even employee strikes also compound the challenges, leading to critical shortages.

In 2020, the BCI report found that a significant number of organizations lacked comprehensive business continuity plans. Among these, a staggering 73% had faced severe disruptions in their supply chains due to the impact of COVID-19, ultimately resulting in their financial collapse [6]. Consequently, a current statistic reveals that 20% of these managers have opted to maintain higher inventory levels to mitigate potential future disruptions. Additionally, 27% are actively enhancing their supplier networks to guarantee the uninterrupted delivery of ordered goods [7]. Very recently, amidst escalating global tensions such as Russia's invasion of Ukraine and China's persistent threat to take over Taiwan by force, Intel Corp. has strategically unveiled plans to invest over USD 50 billion in new semiconductor manufacturing facilities across Poland, Germany, and Israel. Recognizing the precarious geopolitical landscape, Intel is placing its bets on establishing manufacturing plants in multiple countries as a protective measure against potential disruptions [8]. By expanding its operations across these diverse locations, Intel aims to fortify its supply chain resilience. Intel's substantial investments and strategic maneuvering reflect its proactive stance in safeguarding its operations against external disruptions. Typically, these external disruptions are unpredictable yet occasional events that are caused by natural disasters, political instability, labour unrest, industrial accidents, the advent of pandemic diseases, etc. [9]. This study focuses solely on the supply-side-disruption-related crises of cement retailers that result from shortages of input materials, machinery breakdowns, and soaring input costs and continue for a random duration.

On the other hand, the rapidly fluctuating demand rate is among the major concerns of cement retailers [10]. For example, the demand for cement bags improves in the June quarter (i.e., before the rainy season) and slows down during the rainy season owing to sluggish construction activities during the latter. Moreover, several local factors, such as the use of cement in industries, the construction of large urban housing complexes, and elections, greatly affect the demand for cement bags at local retailers. In the case of selling-price-dependent demand, there has also been a significant role of deterioration [11]. Together with fluctuating demand, cement retailers need to consider the probable number of sequential supply failures that lead to a rapid increase in the inventory lead time from zero (the normal business scenario) to multiplications of the review interval (during disruptions) [12]. When supply-side disruptions are prolonged for various reasons, this causes shortages of cement bags at retailers. However, the construction activities of individual home builders and small builders, which account for 60–70% of rural sales and 40–50% of urban sales in India, come to a complete halt without cement. During supply-side disruptions, a segment of customers is unable to wait at a single cement retailer with a fixed cycle length and shifts to

other retailers. This accounts for the partial back-ordering of shortages [13]. Thus, cement retailers need to predict and meet real-time fluctuations in demand and influence them to achieve long-term business goals while ensuring operational agility and resilience to potential market adversities [14]. Accordingly, many retailers employ some sort of artificial intelligence and forecasting-based semi-autonomous decision support (SADS) systems that can predict the demand for cement bags and communicate an upcoming order to cement manufacturers at the start of each review cycle.

Cement is a deteriorating item [15]. Therefore, the leakage and entry of moisture into cement bags during inventory stock-in and/or storage at any retailer lead to an irreversible loss of cement quality, thus making some bags useless. Ghandehari and Dezhtaherian [16] modeled a deteriorated inventory model with partial back-order. During the COVID-19 outbreak, nearly 26% of retailers holding 400+ bags offered major discounts to avoid spoilage. As with any periodic-review inventory system, cement retailers typically discard deteriorated cement bags at the end of each period, incurring some additional costs. These give the base-stock policy a capacity limit to bind the replenishment quantity in each period significant for the planned single-item long-run periodic-review inventory system of cement retailers [17]. Existing sub-optimal base-stock levels lead to a loss of business opportunities for cement retailers with a fixed cycle length under stochastic supply disruptions.

Nevertheless, under probable supply-side disruptions, most existing EOQ models consider specific probability distributions based on exogenous review intervals that sometimes lead to sub-optimal replenishment decisions under longer intervals and increase the risk of inventory obsolescence [18]. This way, to deal with probable supply-side disruptions under a varying demand rate and partial back-ordering of shortages for deteriorating items, the present study frames various cost components, such as expected long-run ordering, acquisition, holding, deterioration, and shortage costs, including back-ordering and lost sales costs, to obtain the retailer's expected long-run net profit of one stochastic periodic-review base-stock inventory model.

Regarding the structure of the rest of this paper, Section 2 deliberates on the background study in terms of two different aspects. Section 3 discusses the notations, assumptions, and problem statements. Next, this study formulates the proposed inventory model of cement retailers in Section 4, while Section 5 analytically establishes the global optimality of the proposed model at the critically determined base-stock level. Later, Section 6 numerically evaluates the proposed model. The 107 managerial insights are derived from the sensitivity analysis of several major parameters here, along 108 with comparisons to some well-established articles with comparisons to some well-established articles. Lastly, Section 8 concludes the paper along with some scopes of future research.

## 2. Literature Review

In the last few decades, well-established inventory articles have deliberated on a number of deterministic and stochastic lot-sizing models through the progressive embedding of diverse real-life scenarios. With effective management decisions being critical to maintaining an efficient and balanced flow of the inventory models Ghasemi et al. [19], the current review focuses on the recent and well-established inventory control and SC models with the following focal points.

### 2.1. Review of Inventory Models under Various Disruptions

The shortcomings of any complicated inventory model, covering products from computer chips to toilet paper, under diverse disruptions, have been evident in recent times. Researchers have discussed various robust and resilient inventory-control strategies by examining the ability of speedy recovery of organizations from a disruptive state to the preceding state and/or a more desirable state, a major aspiration of inventory managers Duchek [20]. Hosseini et al. [21] solved a cost-minimization bi-objective model under a stochastic environment for the allocation of orders and the selection of suppliers. He

showed the alleviation of disruption-related risk while determining the crucial suppliers and optimal order (re)-allocation particulars.

Recently, Alena Puchkova and Thorne [22] addressed the management policies used to reduce the effect of multiple spontaneous disruptions to an EPQ network of an industrial power laboratory's exhibit technique. They resolved the optimal spot and portion in an inventory of protectors associated with other medium nodes. They examined the part and positioning of appraisal areas across the financial exhibit grid to ensure the punctual delivery of superior grade outcomes while observing any faults at the earlier time. Following this, Yavari and Zaker [23] demonstrated the dangers concerning the disturbances of a control network in a resilient form with a closed-loop deteriorated supply chain model. They discovered that integrating two networks resulted in lower cost and more CE.

Supply disruption is a notable crisis in the business environment, and many researchers have become interested in this topic. In a pharmaceutical supply-chain-based model with stochastic demand, Lücker et al. [24] fully characterized some efficient risk management strategies by considering the inventory and reserve capacity strategies together with the mixed and passive acceptance strategies. Their investigation explained how the optimal risk-mitigation strategy depended on the functional characteristics of products and the agile characteristics of the supply chain. Concurrently, in one continuous-review inventory model in the presence of a fully lost sale of unsatisfied demand with constant deterministic demand, Sevgen and Sargut [25] extended an economic order quantity (EOQ) model, in which random disruptions occur on both the supplier and retailer sides. They considered the supply-side disruption in an available and unavailable state. But the retailer was disrupted when all on-hand inventory was destroyed. They identified a cost-saving non-zero reorder point and compared their model with the classical EOQ model. Subsequently, Konstantaras et al. [26] explained the ideal $(S, T)$ base-stock policy minimizing the long-run average cost of an EOQ model under the endogenous supply-side disruptions in an exact analysis with both continuous and end-of-cycle costing schemes. They investigated the impacts of applying any heuristics on the long-run average cost by dodging supply-side disruptions and relying on inaccurate costing information. During the time, in a recurring appraisal of base-stock inventory policy, Saithong and Luong [27] applied a two-phase heuristic algorithm in the existence of supply disruption with a complete backlogging of shortages. They modeled the supply-side disruption span as a continuous random variable that did not affect the lead time. They determined the optimal base-stock level and expected total inventory cost per unit time under the effect of supply disruption. On the other hand, He et al. [28] investigated the optimal ordering decision policies of retailers in a supply-side-disruption-based SC model with correlated demand and price uncertainty. The real-option theory-based model was put in the explicit form of the profit function. Their investigation regarded the applicability of the suggested model in Chinese dairy market companies, e.g., Yili, Mengniu, and Bright.

Unexpected events highly deteriorate the performance of a supply chain. Olivares-Aguila and ElMaraghy [29] investigated the proactive and reactive strategies in disruptions of a multi-echelon supply chain. They examined the inventory model with full and partial disruptions with a consequence on the service levels, costs, profits, and inventory levels. Their analysis demonstrated that the disruptions in the downstream levels had a greater impact with respect to the upstream levels on the SC performance, thus requiring more efforts on the disruption policies for downstream partners. On the other hand, Fattahi et al. [30] considered a novel metric that could measure the expected escalations in disruption-induced costs in the SC during the recovery period. They considered the large number of disruptions by applying the quadratic conic optimization, along with the sample average approximation methods to determine the time and cost of recovery. Their results indicated that the increased capacity was hardly effective in designing any resilient SC. Saithong and Lekhavat [31] formulated the optimal base-stock policy to minimize the total cost per unit time of any supply chain with the partial back-ordering of shortages.

Under the continuous random-variable-type stochastic disruptions, they suggested that retailers surge the base-stock level against any escalations in the disruption frequency or disruption length. Thereafter, in an inventory-management model with supply-side disruptions, Taleizadeh et al. [9] designed two periodic-review optimization models with the base-stock policy and thus determined the corresponding minimum expected long-run total costs. They showed that an appropriate ratio of back-ordering during any supply-side disruptions was effective in cutting costs and reducing the obsolescence of items.

Very recently, due to the global COVID-19 pandemic, manufacturers faced long-term supply disruptions [32]. Chen et al. [33] presented a mixed-integer linear programming model to investigate a disruption recovery strategy of a supply chain system. They considered the life cycle and design-change time of a new product to minimize manufacturer losses after disruptions. At the same time, Khan et al. [34] proposed an inventory model for the period till the first lockdown ended. They looked at consumer behaviors and the state of disruptions to supermarket supply chains in England. This research informed us that supply-side disturbance was more critical than demand-side disruption.

*2.2. Review of Backlogged and Disruptions-Induced Partially Backlogged Inventory Models*

In the existence of deterioration, an EOQ model followed the expiration of products with time. In a competitive business situation, discount facilities play an essential role. Shaikh et al. [35] considered two different EOQ inventory models, namely the inventory model for the zero-ending case and the inventory model for the case of the shortage. In both models, demand depended on the price and the stock level, and the shortages were partially backlogged at a rate with the waiting time for the next arrival. They observed that the inventory model with partial backlogged shortage was more economical from the viewpoint of cost minimization. Concurrently, in a non-instantaneous deteriorating inventory model, Li et al. [36] defined a nonlinear fractional program with joint pricing, replenishment, and preservation technology. They used the waiting-time-dependent partially backlogging rate, price-dependent demand, and time-varying deterioration. They found that investing in preservation technology did not always give an optimal solution. The demand of some industries depends on the selling price and the frequency of advertisement of the product under the financial trade credit policy. Shaikh et al. [35] allowed shortages that were partially backlogged with a varying rate on the duration of waiting time of the subsequent order. Nevertheless, the industry follows a three-parameter Weibull distribution deterioration rate, where, in a continuous review of base-stock policy under the lost sales, Kouki et al. [10] provided a method for both complete and partial rejection cases that calculate the best cost-driven base-stock policy under more general compound Poisson demand processes. They obtained the optimal base-stock level for the full rejection case and the approximate base-stock level under the partial rejection policy.

In a non-deterministic inventory model, Rahman et al. [37] proposed a quantum-behaved particle swarm algorithm for interval-valued deterioration rate items with partially backlogged shortages. They assumed that the demand rate depended upon the selling price. Except for the backlogging parameter, all the parameters were considered interval-valued. They found the optimal cycle span, stock-in duration, initial demand excess, highest deficiency, and complementary optimal intermediate returns. At the same time, using a queuing network model, Kouki et al. [38] investigated a base-stock inventory system for perishable products with lost sales based on available lead time and random demand. They found some significant errors by assuming deterministic or exponential distributions for the lifetime. For maximum profit and minimum risk, multiple-period trade credit plays an essential role in business enterprises. Das et al. [39] established a conserving-technology-based non-instantaneous deteriorated item with partially backlogged multi-period-based trade credit policies with different rates of interest charged. However, the demand rate was variable according to the selling price of the product. However, in the state of a lost sale, Goldberg et al. [40] framed a high-dimensional perishable inventory model. They presented the above model using the asymptotic analysis process for well-approximation. They found

that asymptotic analysis had recently led to significant progress in lost-sales models, dual-sourcing models, and Assemble-to-Order systems in the presence of considerable lead times. In the state of available components, they did not serve a product until their target crossed the backlogging level, and they maximally eliminated all those products that exceeded the backlog target.

Recently, in many firms, backlogging has been a common phenomenon [41]. In the case of unavailability of stock, the consumer chooses two possible paths. In some situations, they wait for the next replenishment, which is treated as a back-order, and those who do not want to wait for the next replenishment will lose sales. In analogous research, Agrawal and Jia [42] formulated a base-stock-policy-based stochastic inventory-management model. The suggested model had a convex asymptotic average cost function under the lost sales and positive lead times with censored demand. They found the expected infinite-horizon average cost to be a convex function in the base-stock level under the non-zero probability for the zero-demand probability distribution. Also, they combined the fixed and known lead time with some unknown demand distribution parameters to develop a learning algorithm with a linearly dependent regret bound. In a sporadic examination of perishable inventory procedures with a specified yield lifetime, Bu et al. [43] included a base-stock policy and partial backlogging. They showed that an easy base-stock model was asymptotically optimal for enlarging any one out-of-yield lifetime, order population extent, unit liability cost, and unit outdating expense.

*2.3. Current Contributions*

In the realm of inventory management, many established EOQ models are built upon specific probability distributions, dictating exogenous review intervals. Unfortunately, these models often yield sub-optimal replenishment strategies and, in certain real-world scenarios, amplify the risk of inventory obsolescence. To address this limitation and embrace the complexities of practical situations, this research sets out to navigate the intricacies of periodically reviewed stochastic demand rates. Moreover, this study accounts for the partial back-ordering of shortages for items experiencing deterioration, all the while considering the backdrop of stochastic disruptions in the supply chain.

The current investigation of this study focuses on a comprehensive evaluation of a periodic-review base-stock inventory model tailored for cement retailers. The crux of this approach lies in dissecting an array of cost components, including ordering costs, acquisition costs, holding costs, deterioration costs, and the expenses incurred due to shortages through both back orders and lost sales. By holistically assessing these components, this study unveils the anticipated long-term net profit.

A distinctive feature of this study is the pursuit of verifying the concavity inherent in the retailer's projected long-term net profit across various operational scenarios. This is anchored in the identification of a critically determined base-stock level—a juncture that profoundly influences the dynamics of inventory control. By substantiating the concavity of the expected net profit, this study enhances the robustness of the proposed model and provides a more comprehensive understanding of the retailer's optimal decision-making landscape.

In essence, this contribution transcends the confines of conventional EOQ models by integrating stochastic demand rates, partial back-ordering for deteriorating items, and supply-side uncertainties. Through a judicious examination of cost components and the validation of profit concavity, this study strives to equip cement retailers with a heightened ability to make informed and strategic inventory-management choices. In doing so, this study mitigates sub-optimal decisions and bolsters resilience in the face of dynamic and uncertain market conditions (Table 1 and Figure 1).

**Table 1.** Comparative review of recent inventory model studies. NA: not applicable.

| Contributors | Model Types | Disruptions | Demand Types | Backlogging |
|---|---|---|---|---|
| Konstantaras et al. [26] | unified cost | supply-side | fixed rate | fully backlogged |
| Saithong and Luong [27] | base-stock | supply-side | uncertain | NA |
| Saithong and Lekhavat [31] | base-stock | supply-side | deterministic | partial |
| Taleizadeh et al. [9] | base-stock | supply-side | deterministic | partial |
| Das et al. [39] | trade credit | NA | selling price dependent | partial |
| Mashud et al. [44] | hybrid payment | supply-side | price-sensitive | partial |
| Bu et al. [43] | base-stock | NA | i.i.d poisson demand | partial |
| Chen et al. [33] | disruption recovery | supply-side | uncertain demand | partial |
| Agrawal and Jia [42] | base-stock | NA | censored demand | lost sale |
| Malmberg and Marklund [45] | base-stock | NA | Poisson demand | fully backlogged |
| Noble et al. [46] | base-stock | NA | uniformly distributed | lost sale |
| Wang [47] | base-stock | NA | retrial demand | partial |
| Wang et al. [48] | base-stock | NA | truncated and convoluted demand | NA |
| Present study | base-stock | supply-side | selling price dependent | partial |

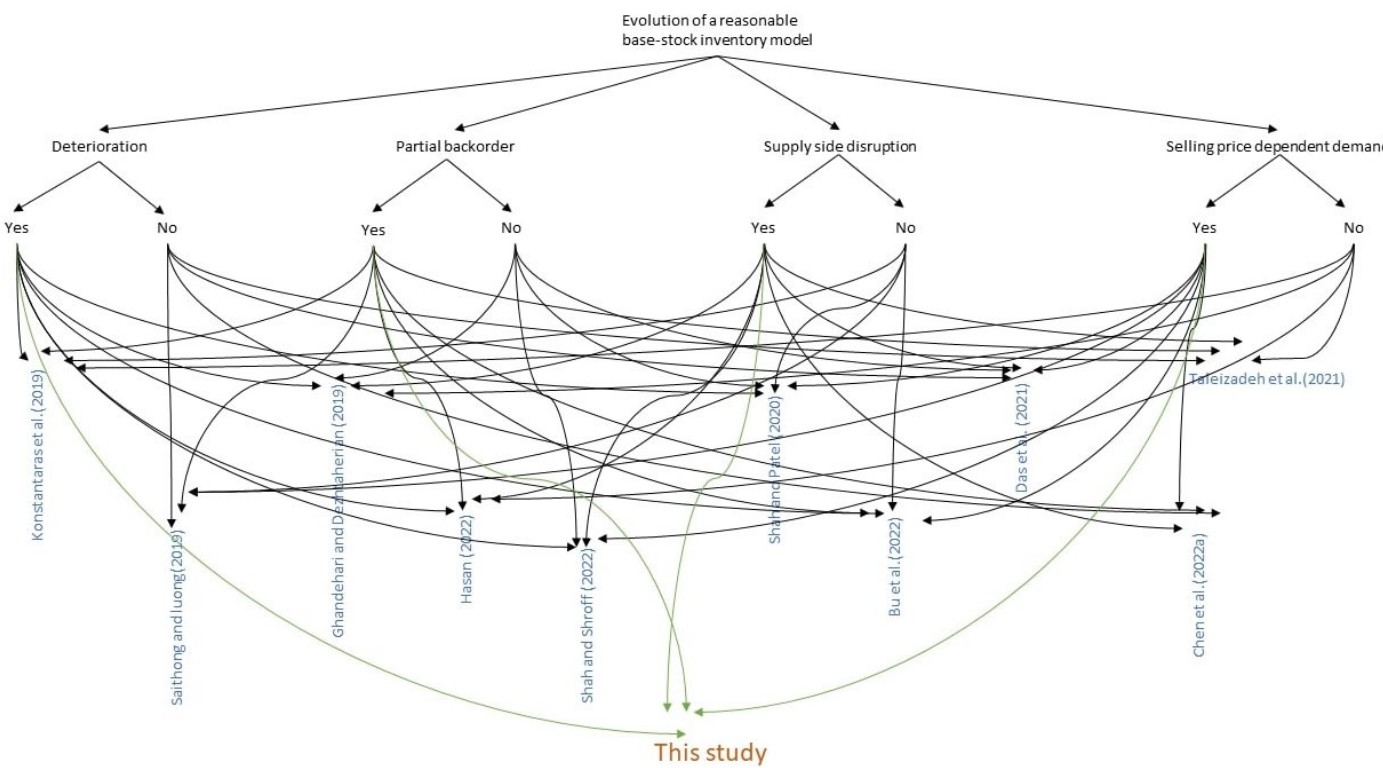

**Figure 1.** A comparative analysis of the existing literature on inventory models [9,11,16,18,26,31,32, 39,41,43].

## 3. Notations, Assumptions, and Problem Statement

### 3.1. Notations

This proposed model uses multiple symbols to explain parameters, variables, and functions. The proper notations are listed below to explain their purpose (Table 2).

**Table 2.** Notations with their descriptions to be used in the proposed model.

| Notations | Descriptions |
|---|---|
| ● Abbreviations | |
| E(.) | indicating the expected values. |
| SADS | semi-autonomatized decision support. |
| EOQ | economic order quantity. |
| ● Decision variables | |
| S | base-stock level or order-up-to level of sale-able products in the inventory of the retailer. |
| T | time interval between two successive reviews with the SADS system of the retailer. |
| (*) | indicates the optimal values. |
| ● Stochastic parameters | |
| X | exponential random variable at rate $\lambda$ acting until the sequential disruptions begins. |
| Y | geometric random variable representing length of sequential disruptions going on for $y$ cycles. |
| Z | random variable representing the length of any renewal cycle, i.e., $Z = X + Y$. |
| $w_1$ | stochastic parameter describing duration of last replenishment before disruption until the arrival of disruption. |
| $w_2$ | the fraction of the cycle length up to which the demand will be fulfilled in the $(m + r)^{th}$ inventory epoch. |
| $w_3$ | the fraction of the cycle length starting from the normalcy of supplier until the next replenishment in the $(m + k)^{th}$ inventory epoch. |
| N | random variable describing the number of full inventory cycles until the arrival of disruptions. |
| m | expectation of the random variable $N$. |
| $D_i$ | sale price dependent random demand rate function to be periodically reviewed by the SADS. |
| $\epsilon_i$ | time-independent zero mean continuous random variable. |
| $\tau$ | state random variable. |
| ● Parameters | |
| $c_h$ | per unit holding cost at the inventory of the retailer. |
| $c_o$ | fixed ordering cost per order of the retailer. |
| $c_p$ | procurement cost per unit item of the retailer from the supplier. |
| $c_b$ | retailer's per unit item back-ordering cost. |
| $c_l$ | retailer's per unit item lost sales cost. |
| $p_i$ | sale price of each item in the $i^{th}$ inventory epoch as determined by the SADS of the retailer. |
| $p_{s_i}$ | scrap price of each deteriorated item in $i^{th}$ inventory epoch to be determined by the SADS. |
| $p$ | the probability of supply-side disruption. |
| $\alpha$ | fixed deterioration rate of the items to be stored in inventory of retailer ($0 \leq \alpha \leq 1$). |
| $\beta$ | percentage of maximum expected shortages to be backlogged at the retailer ($0 \leq \beta \leq 1$). |
| $\mu$ | the probability of reporting by the supplier that the disruption is over. |
| $\lambda$ | arrival rate of sequential disruptions in the supply of sale-able products. |
| $k + 1$ | number of inventory cycles without any replenishment owing to the sequential disruptions. |
| $w_1$ | fraction of the cycle length after which the disruption begins with the supplier. |
| $w_2$ | fraction of cycle length in retailer's last on-hand inventory cycle during disruptions. |

*3.2. Assumptions*

- Retailers aim to keep the competitive advantages through efficient responsiveness, while they are highly interested in the critical responses of customers during stock-outs. Therefore, even though the flaring holding cost of cement bags for retailers is a matter of concern, the stock-out-related retaliations of customers under supply-side disruptions far outweigh these, leading to adaptation of the base-stock policy in the proposed model [9].

- This study estimates the probability of independent supply-side disruptions to occur any number of times consecutively. A random variable $\tau$, with the following geometric distribution, represents the number of delivery failures of the proposed inventory system [9]:

$$\pi_\tau = P(X = \tau) = p^\tau(1 - p),\ \tau = 0,\ 1,\ \ldots \tag{1}$$

Thus, in a regular span (i.e., without disruption), $\tau = 0$.

- On the basis of the market-determined sale price in the proposed single-product periodic-review inventory system, SADS shall place an optimally determined order quantity for cement bags at the beginning of each inventory cycle, thus countering any possible price-sensitive demand rate in the current cycle. This study takes the autonomically predicted random demand rate of retailers in the following additive form:

$$D_i(P_i) = d_i(P_i) + \epsilon_i,\ i = 1, 2, \ldots \tag{2}$$

where $d_i(P_i) \in [\underline{d}_i, \bar{d}_i] > 0, i = 1, 2, \ldots$ is one deterministic and strictly monotonically decreasing function in $P_i \in [\underline{p}_i, \bar{p}_i], i = 1, 2, \ldots$ This ensures that $d_i$ is non-negative at all times and its inverse is a continuous, differentiable, and strictly decreasing function. Also, $\epsilon_i,\ i = 1, 2,\ \ldots$ describes a time-independent, hardly identical, zero-mean continuous random variable with the probability density function $\psi_i(.)$. Here, the minimum price to charge is more than or equal to the discounted ordering cost of the next cycle.

- Any unsatisfied demand during prolonged supply-side disruptions is partially back-logged and is assumed to be fixed for the sake of simplicity alone [9]. Except for the phases of potential supply-side disruptions, the shortages do not occur, and the stock level to be determined using SADS is sufficient to meet any demand rate.

- The procurement lead time is promised to be nil at the time of the contract between the supplier and retailer. During disturbances, however, the lead time rises from 0 to T, 2T, 3T, $\ldots$ based on the likelihood of the random variable returning values $1, 2, \ldots$.

- For the sake of simplicity alone, this study considers that all stocks deteriorate at the same pace, regardless of their lifetime.

*3.3. Communicative Problem Statement*

This study plans to determine the optimal base-stock inventory and associated strategies for cement retailers facing randomly varying demand rates and partially back-ordered shortages under stochastic supply-side disruptions. Furthermore, this study investigates how prolonged disruptions, fluctuating cost components, and other related scenarios influence cement retailers' optimal base-stock levels and corresponding long-run profitability.

## 4. Formulation of the Proposed Model

The present section designs an inventory control model to measure the optimal order up to the level that maximizes the total expected long-run net profit of the retailers. In practice, a number of retailers selling deteriorating items, like Nestlé S.A., BRF S.A., and Cargill, find the stock-out-related responsiveness to be more critical than the holding cost imposed on them, thereby keeping the competitive advantages through responsiveness at an ideal level. Accordingly, this study determines the different long-run expected

cost components, namely the ordering cost, the acquisition cost, the holding cost, the deterioration cost, the shortage cost, and the ost sales cost, within both the regular span and the disruptive spans, as illustrated as follows:

### 4.1. Cost Components in the Regular Span

**Expected long-run ordering cost**

The retailer bears the one-time fixed ordering cost while placing the replenishment order at the beginning of any inventory cycle during the regular span. Thus, subject to unit ordering cost $c_o$, this study determines the expected long-run ordering cost of the retailer within the regular span as follows:

$$OC_{NDC} = c_o(m-1). \tag{3}$$

**Expected long-run holding cost**

A suitable place, store, warehouse etc., is essential to store the saleable products. While the resulting holding cost includes the cost of insurance, numerous taxes, maintenance costs, electricity costs, and many others, researchers find this cost to be proportional to the number of products in the inventory within the holding period. Here, the area of the rectangle in any $i^{th}, i = 1, \ldots, (m-1)$ inventory epoch (see, for details, Figure 2) is $R_i = (S - D_i T)T, i = 1, \ldots, (m-1)$, while the corresponding area of a triangle is $\triangle_i = \frac{D_i T^2}{2}, i = 1, \ldots, (m-1)$. This provides the total area under the inventory curve in any $i^{th}, i = 1, \ldots, (m-1)$ inventory epoch within the regular span as follows:

$$A_i = \frac{D_i T^2}{2} + (S - D_i T)T, i = 1, \ldots, (m-1). \tag{4}$$

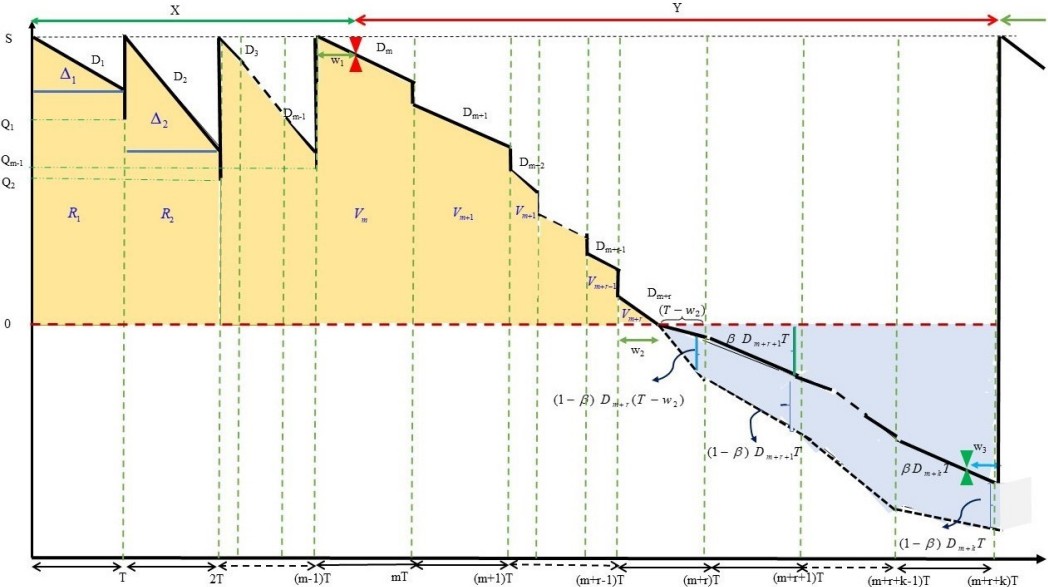

**Figure 2.** Graphical illustration of the resilient inventory profile with partial back-ordering.

Therefore, the expected long-run holding cost of the retailer within the regular span is as follows:

$$HC_{NDC} = c_h \sum_{i=1}^{m-1} A_i = c_h \sum_{i=1}^{m-1} \left( \frac{1}{2} \left( D_i T^2 \right) + (S - D_i T)T \right). \tag{5}$$

**Expected long-run deterioration cost**

The retailer discards the products that deteriorate over time from its inventory at the end of each on-hand inventory cycle. Thus, this study considers the expected long-run deterioration cost of the retailer within the regular span as follows:

$$PC_{NDC} = \alpha \sum_{i=1}^{m-1} (p_i - p_{s_i}) A_i = \alpha \sum_{i=1}^{m-1} (p_i - p_{s_i}) \left( \frac{1}{2} \left( D_i T^2 \right) + (S - D_i T) T \right). \quad (6)$$

**Expected long-run shortage cost**

Since this study assumes that the retailer stores enough stock to fulfill the demand of customers in any inventory cycle during the regular span, the scenario of shortages does not occur during the regular span. Thus, the resulting expected long-run shortage cost is nil.

**Expected long-run acquisition cost**

In any $i^{th}, i = 1, \ldots, (m-1)$ regular inventory epoch, the retailer makes replenishment decisions for inventory up to the maximum stock level S by adding the number of sold items in the previous cycle and the number of deteriorated items to be discarded from inventory at the end of the current cycle. Thus, this study describes the order quantity $Q_i$, for which the SADS of the retailer autonomically places the order, as follows:

$$Q_i = D_i T + \alpha \left( \frac{1}{2} \left( D_i T^2 \right) + (S - D_i T) T \right), i = 1, \ldots, (m-1). \quad (7)$$

In this way, this study exemplifies the long-term expected acquisition cost of retailers within the regular span as follows:

$$AC_{NDC} = c_p \sum_{i=1}^{m-1} Q_i = c_p \left( \sum_{i=1}^{m-1} (D_i T + \alpha(\frac{1}{2} D_i T^2 + (S - D_i T) T) \right). \quad (8)$$

In addition, this is to note that the random variables $X$ and $m$ are connected through the following relationship:

$$P\{N = n\} = P\{(n-1)T \leq X \leq nT\} \text{ and } E(N) = \sum_{n=1}^{\infty} nP\{N = n\}, \quad (9)$$

$$\text{i.e., } E(N) = \frac{1}{1 - e^{-\lambda T}} = m, \text{say.}$$

*4.2. Cost Components in the Disruptive Span*

With the aim to efficiently manage the sequential disruptions occurring randomly to the supplier in the $m^{th}$ inventory cycle and staying for the random duration $Y$ (i.e., $f_y(y) = \mu(1-\mu)^{y-1}$), this study assumes the inventory state random variable $\tau$ to range from the $m^{th}$ to at most the $(m+r)^{th}$ on-hand inventory cycle under the disruptions. Whenever the successive disruptions continue to occur till the $(m+r)^{th}$ cycle or more, the retailer experiences shortages. Consequently, the supplier makes the replenishment afresh at the end of the $(m+\tau)^{th}, \tau = 0, \ldots, (r+k+1)$ inventory cycle.

**Expected long-run ordering cost**

The retailer does not place any request for the replenishment of stocks to the supplier during the disruptions but asks for the one-time replenishment only after the supply-side disruptions are over. Thus, this study obtains the expected long-run ordering cost of the retailer during supply-side disruption cycles as follows:

$$OC_{DC} = c_o. \quad (10)$$

**Expected long-run holding cost**

This study measures the expected long-run holding cost and the associated deterioration cost by determining the areas of the various shaded regions displayed in Figure 2. Here, the $m^{th}$ inventory epoch follows the last replenishment before the sequential disruptions occur. Thus, the retailer's $m^{th}$ inventory cycle begins with $S$ units of items in its inventory during the disruptive span. Therefore, this study computes the area $V_m$ of the trapezoidal region under the inventory curve at the $m^{th}$ inventory epoch within the disruptive span as follows:

$$V_m = \frac{(S + (S - D_m T))T}{2} = \left(S - \frac{D_m T}{2}\right)T. \tag{11}$$

While the retailer discards the deteriorated items at the end of each on-hand inventory cycle during the disruptive span, this study determines the area of the trapezoidal region $V_{m+1}$ under the inventory curve at the $(m+1)^{th}$ on-hand inventory epoch within the disruptive span as follows:

$$V_{m+1} = \left(S - D_m T - \alpha V_m - \frac{D_{m+1} T}{2}\right)T. \tag{12}$$

Therefore, this study computes the area of the trapezoid representing the $(m+r-1)^{th}$ cycle as follows:

$$V_{m+r-1} = \left(S - T \sum_{i=m}^{m+r-2} D_i - \alpha \sum_{i=m}^{m+r-2} V_i - \frac{D_{m+r-1} T}{2}\right)T. \tag{13}$$

All these yield the area of the trapezoid representing any $\tau^{th}, 0 \leq \tau \leq (r-1)$, cycle as follows:

$$V_{m+\tau} = \left(S - T \sum_{i=m}^{m+\tau-1} D_i - \alpha \sum_{i=m}^{m+\tau-1} V_i - \frac{D_{m+\tau} T}{2}\right)T. \tag{14}$$

Whereas Figure 2 suggests that the range of the on-hand inventory state random variable ($\tau$) extends from $m$ to $(m+r)$ during the disruptive span, this study conceives that the retailer is capable of satisfying the demand of customers solely for the duration $w_2$ ($0 < w_2 \leq T$) in the $(m+r)^{th}$ inventory epoch during the disruptive span. Accordingly, the area $V_{m+r}$ of the triangle in the on-hand inventory at the $(m+r)^{th}$ inventory epoch is as follows:

$$V_{m+r} = \frac{D_{(m+r)} w_2^2}{2},$$

subject to the fraction $w_2$ of the on-hand inventory period in the $(m+r)^{th}$ inventory epoch, which is as follows:

$$
\begin{aligned}
w_2 &= \frac{S - \sum_{i=m}^{m+r-1}(D_i T + \alpha V_i)}{D_{m+r}} \\
&= \frac{S - \sum_{i=m}^{m+r-1} D_i T - \alpha f_1(S)}{D_{m+r}}
\end{aligned}
\tag{15}
$$

see, for details, Appendix C.

Thus, this study determines the area of the region of any $\tau^{th}, \tau = 0, \ldots, (m + r)$ inventory epoch as follows:

$$I_{m+\tau} = \begin{cases} V_{m+\tau}, \text{if } 0 \le \tau \le (r-1) \\ V_{m+r}, \text{if } \tau = r. \end{cases} \tag{16}$$

The long-run average inventory of the retailer in the disruptive span is as follows:

$$E(I_{m+\tau}) = \sum_{\tau=0}^{\infty} V_{m+\tau} \pi_\tau = \sum_{\tau=0}^{r-1} V_{m+\tau} \pi_\tau + V_{m+r} \pi_r. \tag{17}$$

In this way, subject to the per unit holding cost $c_h$, this study measures the long-term expected holding cost of the retailer in the disruptive span as follows (see, for details, Appendix D):

$$HC_{DC} = c_h E(I_{m+\tau})$$
$$= c_h \left( \sum_{\tau=0}^{r-1} V_{m+\tau} \pi_\tau + V_{m+r} \pi_r \right) = c_h \left( f_2(S) + (1-p)p^r \frac{(S - \sum_{i=m}^{m+r-1} D_i T - \alpha f_1(S))^2}{2D_{m+r}} \right). \tag{18}$$

**Expected long-run deterioration cost**

Subject to the average on-hand inventory in the disruptive span as obtained in the relation (13), the uniform deterioration rate $\alpha$, and the unit scrap price $p_{s_i}, i = m, \ldots (m + r - 1)$ in any $i^{th}$ inventory epoch, this study computes the long-term expected deterioration cost ($PC_{DC}$) of the retailer in the disruptive span as follows:

$$PC_{DC} = \sum_{i=m}^{m+r-1} (p_i - p_{s_i}) \alpha E(I_\tau)$$
$$= \sum_{i=m}^{m+r-1} (p_i - p_{s_i}) \alpha \left( \sum_{\tau=0}^{r-1} V_{m+\tau} \pi_\tau + V_{m+r} \pi_r \right) \tag{19}$$
$$= \sum_{i=m}^{m+r-1} \left( (p_i - p_{s_i}) \alpha \left( f_2(S) + (1-p)p^r \frac{(S - D_i T - \alpha f_1(S))^2}{2D_{m+r}} \right) \right).$$

Negative Inventory Period

It is highly unpredictable to specify the period for which the sequential disruptions will keep going. Nevertheless, pessimistically, the shortages start to happen after the sequential disruptions reach the $(m + r)^{th}$ inventory cycle. By considering $w_2$ to be the fraction of the cycle length for which the demand is fulfilled from the inventory of the retailer, this study measures the expected long-run shortage cost consisting of the expected long-run back-ordering cost and the expected long-run lost sales cost as follows:

**Expected long-run back-ordering cost**

Here, the area of the right-angled triangular region $A_1 A_2 A_3$ (see Figure 3) representing the partially back-ordered inventory for the duration $(T - w_2)$ in the $r^{th}$ inventory cycle post disruption (i.e., $(m + r)^{th}$ inventory epoch) is as follows (see Appendix B).

Here, the sides of the triangle $A_1 A_2 A_3$ are expressed as $A_1 A_2 = (T - w_2)$ and $A_2 A_3 = \beta D_{m+r}(T - w_2)$.

$$W_{m+r} = \frac{\beta D_{m+r}(T - w_2)^2}{2}. \tag{20}$$

Likewise, this study measures the area of the trapezoidal region $A_2 A_5 A_7 A_3$ describing the partially back-ordered inventory within the $(m + r + 1)^{th}$ inventory cycle as follows.

Here, the area of the trapezoidal region $A_2A_5A_7A_3$ is obtained as the sum of two regions, one rectangular region $A_2A_5A_6A_3$ with sides $A_2A_5 = T$ and $A_5A_6 = \beta D_{m+r}(T - w_2)$, the triangular region with sides $A_3A_6 = T$, and $A_6A_7 = \beta D_{m+r+1}T$ (see Figure 3).

$$W_{m+r+1} = \frac{\beta D_{m+r+1}T^2}{2} + \beta D_{m+r}(T - w_2)T. \tag{21}$$

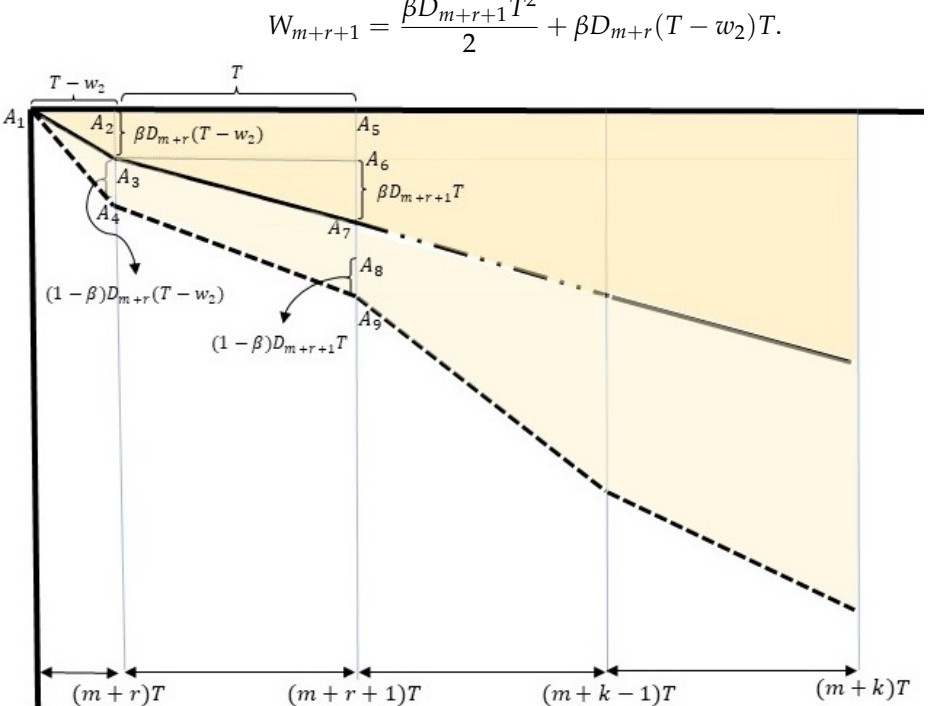

**Figure 3.** Graphical illustration of the negative inventory period.

In this way, the present study measures the area of the trapezoidal region of the partially back-ordered inventory in any $(m + \tau)^{th}$, $\tau \geq r + 1$ inventory cycle as follows:

$$W_{m+\tau} = \left( \beta D_{m+r}(T - w_2) + \sum_{i=r+1}^{\tau-1} \beta D_{m+i}T + \frac{\beta D_{m+\tau}T}{2} \right)T, \ \tau \geq r + 1. \tag{22}$$

Analogous to the deliberation on the long-run average inventory of the retailer, this study computes the average back-ordering at the retailer as follows:

$$J_{m+\tau} = \begin{cases} W_{m+r}, \ if \tau = r, \\ W_{m+\tau}, \ if \tau \geq r + 1. \end{cases} \tag{23}$$

Thus, the expected average back-ordering at the retailer is as follows:

$$E(J_{m+\tau}) = \sum_{\tau=r}^{\infty} \pi_\tau W_{m+\tau} = \left( \pi_r W_{m+r} + \sum_{\tau=r+1}^{\infty} \pi_\tau W_{m+\tau} \right). \tag{24}$$

The following equations provide the expected long-run back-ordering cost of the retailer:

$$BC_{DC} = c_b E(J_{m+\tau})$$

$$= c_b\left(\frac{1}{2}(1-p)p^r \beta D_{m+r}\left( T - \frac{S - \sum_{i=m}^{m+r-1} D_i T - \alpha f_1(S)}{D_{m+r}} \right)^2 + \frac{1}{2}\beta T^2 \sum_{\tau=r+1}^{\infty} D_{m+\tau} \pi_\tau \tag{25}$$

$$+ p^{r+1}\beta D_{m+r}\left( T - \frac{S - \sum_{i=m}^{m+r-1} D_i T - \alpha f_1(S)}{D_{m+r}} \right)T + \beta T^2 \sum_{\tau=r+2}^{\infty} \pi_\tau \sum_{i=r+1}^{\tau-1} D_{m+i} \right).$$

**Expected long-run lost sales cost**

This study measures the lost sales vertically by computing the difference between the actual inventory position and its position at $\beta = 1$ at the end of any negative inventory cycle (see Figure 3), thereby deducing it as follows.

Here, the amount of lost sales for the $(m+r)^{th}$ epoch is expressed as the length $A_3 A_4 = (1-\beta)D_{m+r}(T-w_2)$. For the $(m+r+1)^{th}$ epoch, the lost sales are represented by the segment $A_7 A_9$, which is the sum of the length $A_7 A_8 = A_3 A_4 = (1-\beta)D_{m+r}(T-w_2)$ and $A_8 A_9 = (1-\beta)D_{m+r+1}T$. Therefore, $A_7 A_9 = (1-\beta)D_{m+r}(T-w_2) + (1-\beta)D_{m+r+1}T$. Likewise, the present study measures the amount of lost sales in any $(m+\tau)^{th}$, $\tau \geq r+1$ cycles (see Equation (26)).

$$L_{m+\tau} = \begin{cases} (1-\beta)D_{m+r}(T-w_2), & \text{if } \tau = r, \\ L_{m+r} + \sum_{i=r+1}^{\tau}(1-\beta)D_{m+i}T, & \text{if } \tau > r. \end{cases} \tag{26}$$

Thus, this study finds the following:

$$E(L_\tau) = \sum_{\tau=r}^{\infty} \pi_\tau L_{m+\tau} = \pi_r L_{m+r} + \sum_{\tau=r+1}^{\infty} \pi_\tau L_{m+\tau}. \tag{27}$$

In this way, this study computes the expected average lost sales cost of the retailer as follows:

$$LS_{DC} = c_l\left(\pi_r L_{m+r} + \sum_{\tau=r+1}^{\infty} \pi_\tau L_{m+\tau}\right)$$

$$= c_l\left[(1-\beta)p^r D_{m+r}\left(T - \frac{S - \sum_{i=m}^{m+r-1} D_i T - \alpha f_1(S)}{D_{m+r}}\right) + T\sum_{\tau=r+1}^{\infty}\pi_\tau\sum_{i=r+1}^{\tau}(1-\beta)D_{m+i}\right]. \tag{28}$$

**Expected long-run acquisition cost**

The base-stock level $S$ and the total amount of back-ordered portion in any $i^{th}$, $i = (m+r), (m+r+1), \ldots, (m+k)$ inventory epoch during disruption is described in Figure 2 as $W_{m+\tau}, \tau = r, r+1, \ldots, k$, which have to be ordered after the completion of the disruption. Thus, after the completion of the disruption, the SADS will place an order to the supplier. Consequently, the expected acquisition cost of the retailer within the disruptive span is as follows:

$$AC_{DC} = c_p\left(\sum_{\tau=0}^{r-1}\frac{D_{m+\tau}T^2}{2} + \frac{D_{m+r}w_2^2}{2} + \alpha\sum_{\tau=0}^{r-1}V_{m+\tau}\pi_\tau + \sum_{\tau=r}^{\infty}\pi_\tau W_{m+\tau}\right)$$

$$= c_p\left(\sum_{\tau=0}^{r-1}\frac{D_{m+\tau}T^2}{2} + V_{m+r} + \alpha\sum_{\tau=0}^{r-1}V_{m+\tau}\pi_\tau + \pi_r W_{m+r} + \sum_{\tau=r+1}^{\infty}\pi_\tau W_{m+\tau}\right)$$

$$= c_p\left(\sum_{\tau=0}^{r-1}\frac{D_{m+\tau}T^2}{2} + \frac{(S - \sum_{i=m}^{m+r-1} D_i T - \alpha f_1(S))^2}{2D_{m+r}} + \alpha f_2(S) + \right. \tag{29}$$

$$\frac{1}{2}(1-p)p^r\beta D_{m+r}\left(T - \frac{S - \sum_{i=m}^{m+r-1} D_i T - \alpha f_1(S)}{D_{m+r}}\right)^2 + \frac{1}{2}T^2\sum_{\tau=r+1}^{\infty}\beta D_{m+\tau}\pi_\tau + $$

$$\left. \beta D_{m+r}T\left(T - \frac{S - \sum_{i=m}^{m+r-1} D_i T - \alpha f_1(S)}{D_{m+r}}\right)p^{r+1} + T^2\sum_{\tau=r+2}^{\infty}\pi_\tau\sum_{i=r+1}^{\tau-1}\beta D_{m+i}\right).$$

*4.3. Retailer's Expected Long-Run Net Profit per Unit Time*

The amount of saleable items on $i^{th}$ cycle is expressed as $D_i T$, where the selling price is $p_i$, $i = 1, 2, \ldots$.

The total earnings before disruption are expressed as follows:

$$E(E_{NDC}) = \sum_{i=1}^{m} p_i D_i T. \tag{30}$$

The total earnings after disruption are the sum of sellable items with a positive inventory level with the back-ordered portion, expressed as follows:

$$E(E_{DC}) = \left( \sum_{\tau=1}^{r-1} \pi_\tau p_{m+\tau} D_{m+\tau} T + \pi_r p_{m+r} D_{m+r} w_2 \right) +$$
$$\left( \beta \pi_r D_{m+r} (T - w_2) p_{m+r} + \beta \sum_{\tau=r+1}^{\infty} \pi_\tau D_{m+\tau} p_{m+\tau} T \right). \tag{31}$$

This study measures the retailer's expected long-run aggregate earnings (by combining Equations (30) and (31)) per cycle as follows:

$$E(AE) = \left( \sum_{i=1}^{m} p_i D_i T + \sum_{i=1}^{r-1} \pi_i p_{m+i} D_{m+i} T + \pi_r p_{m+r} D_{m+r} w_2 + \pi_r D_{m+r} (T - w_2) p_{m+r} \beta + \right. \tag{32}$$
$$\left. \sum_{i=r+1}^{\infty} \pi_i \beta D_{m+i} p_{m+i} T \right).$$

Again, on the basis of the various cost components of retailers in both the non-disrupted and disrupted cycles (see relation (3) for $OC_{NDC}$, relation (5) for $HC_{NDC}$, relation (6) for $PC_{NDC}$, relation (8) for $AC_{NDC}$, relation (10) for $OC_{DC}$, relation (18) for $HC_{DC}$, relation (19) for $PC_{DC}$, relation (25) for $BC_{DC}$, relation (28) for $LS_{DC}$, and relation (29) for $AC_{DC}$), this study expresses retailer's expected long-run aggregate cost as follows:

$$E(TC(S)) = (OC_{NDC} + HC_{NDC} + PC_{NDC} + AC_{NDC}) +$$
$$(OC_{DC} + HC_{DC} + PC_{DC} + BC_{DC} + LS_{DC} + AC_{DC}). \tag{33}$$

Therefore, this study represents the expected long-run net profit per cycle of the retailer as follows (see, for the full expression, Appendix E):

$$E(NP) = \frac{1}{T}(E(AE) - E(TC)). \tag{34}$$

## 5. Analytical Derivation

The present section analytically establishes the global optimality of retailers' expected long-run net profit using the classical optimization approach. To reduce to essentials regarding the expected long-run net profit per cycle, this study performs the rest of the analysis with two full inventory cycles, along with one on-hand inventory cycle and two cycles with shortages post-disruption. Here, the proposed inventory model is a nonlinear programming problem with the optimal base-stock level decision to be determined.

**Lemma 1.** *Under any circumstances, the expected long-run net profit of the retailer of the proposed inventory model is concave in shape for the base-stock level and globally attains the maximum value at that critical order up to that level.*

**Proof.** This study redrafts the relation (34) to obtain the expected long-run net profit per cycle of the retailer as follows:

$$E(NP(S)) = \frac{1}{T}\left(-c_b\left(\beta D_4(1-p)p^3T^2 + \beta D_3 p^2 T\left(T - \frac{-\alpha\left(ST - \frac{D_2 T^2}{2}\right) - D_2 T + S}{D_3}\right) + \right.\right.$$

$$\frac{1}{2}\beta(1-p)T^2\left(D_5 p^3 + D_4 p^2\right) + \frac{1}{2}\beta D_3(1-p)p\left(T - \frac{-\alpha\left(ST - \frac{D_2 T^2}{2}\right) - D_2 T + S}{D_3}\right)^2\right) - $$

$$\left(ST - \frac{D_1 T^2}{2}\right)(c_h + \alpha(p_1 - p_{s_1})) - c_h\left(\frac{(1-p)p\left(-\alpha\left(ST - \frac{D_2 T^2}{2}\right) - D_2 T + S\right)^2}{2D_3} + \right.$$

$$(1-p)\left(ST - \frac{D_2 T^2}{2}\right)\right) - (1-\beta)c_l\left(T\left(D_5 p^3 + D_4 p^2\right) + \right.$$

$$D_3 p\left(T - \frac{-\alpha\left(ST - \frac{D_2 T^2}{2}\right) - D_2 T + S}{D_3}\right)\right) - $$

$$c_p\left(\beta D_3(1-p)p^2 T\left(T - \frac{-\alpha\left(ST - \frac{D_2 T^2}{2}\right) - D_2 T + S}{D_3}\right) + \beta T^2(1-p)\left(D_5 p^4 + D_4 p^3\right) + \right. \tag{35}$$

$$\beta(1-p)T^2\left(D_5 p^3 + D_4 p^2\right) + \frac{1}{2}\beta D_3(1-p)p\left(T - \frac{-\alpha\left(ST - \frac{D_2 T^2}{2}\right) - D_2 T + S}{D_3}\right)^2 + $$

$$\alpha(1-p)\left(ST - \frac{D_2 T^2}{2}\right) + \frac{\left(-\alpha\left(ST - \frac{D_2 T^2}{2}\right) - D_2 T + S\right)^2}{2D_3} + \frac{D_2 T^2}{2}\right) - $$

$$c_p\left(\alpha\left(ST - \frac{D_1 T^2}{2}\right) + D_1 T\right) - 2c_o + \beta(1-p)T\left(D_5 p_5 p^3 + D_4 p_4 p^2\right) - $$

$$\alpha(p_2 - p_{s_2})\left(\frac{(1-p)p\left(-\alpha\left(ST - \frac{D_2 T^2}{2}\right) - D_2 T + S\right)^2}{2D_3} + (1-p)\left(ST - \frac{D_2 T^2}{2}\right)\right) + $$

$$\beta D_3(1-p)p p_3\left(T - \frac{-\alpha\left(ST - \frac{D_2 T^2}{2}\right) - D_2 T + S}{D_3}\right) + $$

$$(1-p)p p_3\left(-\alpha\left(ST - \frac{D_2 T^2}{2}\right) - D_2 T + S\right) + D_1 p_1 T + D_2 p_2 T\right).$$

To establish the concavity of $E(NP(S))$, this study computes the first-order derivative of $E(NP(S))$ with respect to $S$ to obtain the following relation:

$$\frac{dE(NP(S))}{dS} = -\frac{1}{T}\left(\frac{\beta p c_b(\alpha T - 1)(2(D_3 T + (1 - p)S(\alpha T - 1)) + D_2(p - 1)T(\alpha T - 2))}{2D_3} + \right.$$

$$(1 - p)c_h\left(T - \frac{p(\alpha T - 1)(D_2 T(\alpha T - 2) - 2S(\alpha T - 1))}{2D_3}\right) -$$

$$\frac{1}{2D_3}\left(c_p\left(2\left(D_3 T(\beta p + \alpha(\beta p(-T) + p - 1)) + S\left(\beta p^2 - \beta p - 1\right)(\alpha T - 1)^2\right) - \right.\right.$$

$$\left.D_2 T\left(\beta p^2 - \beta p - 1\right)\left(\alpha^2 T^2 - 3\alpha T + 2\right)\right)\right) + \tag{36}$$

$$T(c_h + \alpha(p_1 - p_{s_1})) + (\beta - 1)(-p)c_l(\alpha T - 1) + \alpha T c_p +$$

$$\alpha(1 - p)(p_2 - p_{s_2})\left(T - \frac{p(\alpha T - 1)(D_2 T(\alpha T - 2) - 2S(\alpha T - 1))}{2D_3}\right) +$$

$$\beta(p - 1)pp_3(\alpha T - 1) - (p - 1)pp_3(\alpha T - 1)).$$

Next, under the necessary condition for the optimality of the unconstrained model, this study equates $\frac{dE(NP(S))}{dS}$ to zero. Thus, this determines the critical value of the base-stock level $S^*$ as follows:

$$S^* = \frac{1}{2(\alpha T - 1)^2\left(\beta(p - 1)pc_b + (p - 1)pc_h + c_p(\beta p^2 - \beta p - 1) + \alpha(p - 1)p(p_2 - p_{s_2})\right)} \times$$

$$\left(D_3 T\left(\frac{\beta D_2(p - 1)pc_b(\alpha T - 2)(\alpha T - 1)}{D_3} + 2\beta pc_b(\alpha T - 1) + \frac{D_2(p - 1)pc_h(\alpha T - 2)(\alpha T - 1)}{D_3} + \right.\right.$$

$$\frac{D_2 c_p\left(\beta p^2 - \beta p - 1\right)\left(\alpha^2 T^2 - 3\alpha T + 2\right)}{D_3} - 2(p - 1)c_h + 2c_h - \frac{2(\beta - 1)pc_l(\alpha T - 1)}{T} + 2\alpha c_p + \tag{37}$$

$$2c_p(\alpha + \alpha(-p) - \beta p + \alpha\beta pT) + \frac{\alpha D_2(p - 1)p(p_2 - p_{s_2})(\alpha T - 2)(\alpha T - 1)}{D_3} + 2\alpha(p_1 - p_{s_1}) -$$

$$\left.\left.2\alpha(p - 1)(p_2 - p_{s_2}) + \frac{2\beta(p - 1)pp_3(\alpha T - 1)}{T} - \frac{2(p - 1)pp_3(\alpha T - 1)}{T}\right)\right).$$

To show the sufficiency part, this study computes the second-order derivative of $E(NP(S))$ with respect to $S$ at said critical point $S^*$, resulting in the following expression:

$$\frac{d^2 E(NP(S))}{dS^2} =$$

$$-\frac{(\alpha T - 1)^2\left(\beta(1 - p)pc_b + (1 - p)pc_h + \beta(1 - p)c_p + c_p + \alpha(1 - p)(p_2 - p_{s_2})p^2\right)}{D_3 T} < 0. \tag{38}$$

This expression is less than zero and independent of $S^*$ under any circumstances. The proof is thus complete. □

*Derivations*

On the basis of the analysis, this study makes the following deductions (see the description of symbols in Appendix G):

- Back-ordered portions in a base-stock inventory model impact the operational efficiency and customer satisfaction. They signify unmet demand due to stock-outs, potentially leading to lost sales and customer dissatisfaction. Balancing back orders optimally helps minimize excess holding costs while meeting demand. The proper management of back orders ensures smoother supply chain operations and enhances customer loyalty by fulfilling orders promptly and efficiently. Hence, it is crucial to establish the correlation between the optimal base-stock level and the back-ordered portion in a cycle, as inferred from the condition $\frac{dE(NP(S))}{dS} = 0$, as shown below:

$$S = \frac{B_1 + B_2}{\beta B_3 + 1} + B_1. \tag{39}$$

Thus, $S \propto \frac{1}{\beta}$. The relationship between the base-stock level and the back-order portion is such that an increase in the back-order portion tends to decrease the base-stock level.

- Lost sales cost ($c_l$) within inventory management involves assessing the expenses incurred per unit of an item that cannot be fulfilled through back orders. In instances of negative inventory, some customers opt to purchase from alternative sources, leading to lost sales. So lost sales occur. Consequently, the dynamic interplay between the base-stock level and the cost of lost sales holds considerable influence over the model. Thus, establishing a relation between the optimal base-stock level of the proposed model and the corresponding lost sales cost is important. This connection can be derived from the condition $\frac{dE(NP(S))}{dS} = 0$, as elaborated below:

$$S = C_2 c_l + C_1. \tag{40}$$

Therefore, $S \propto c_l$. Hence, the base-stock level is directly proportional to the lost sales cost. As the lost sales cost increases, the base-stock level also increases.

- The acquisition cost ($c_p$) is calculated per unit of purchased cost. The retailer determines replenishment choices for inventory, maintaining it to the maximum stock level $S$. This includes adding the number of items sold in prior cycles and those damaged, earmarked for removal at the current cycle's end. These units must be procured, each incurring a per-unit acquisition cost. Thus, the connection between the acquisition cost and the optimal base stock level holds significance. This connection arises from the condition $\frac{dE(NP(S))}{dS} = 0$, as illustrated below:

$$S = \frac{E_2}{E_3 + c_p E_4} + E_1. \tag{41}$$

Thus, $S \propto \frac{1}{c_p}$. Hence, an inverse relationship exists between the base-stock level and the acquisition cost: when the acquisition cost rises, the base-stock level tends to decrease.

- The holding cost ($c_h$) is the per-unit value of the carrying cost. It represents the expenses and financial implications a business incurs for storing and maintaining its inventory over a certain period. Companies aim to balance holding enough inventory to meet customer demand while minimizing carrying costs. Holding costs play a crucial role in determining a business's overall cost structure and profitability. So the relationship between holding cost and base-stock level is essential and is expressed from the state $\frac{dE(NP(S))}{dS} = 0$, as elaborated below:

$$S = \frac{F_2}{F_4 c_h + F_3} + F_1. \tag{42}$$

Thus, $S \propto \frac{1}{c_h}$. As holding costs increase, the base-stock level decreases, and vice versa.

## 6. Numerical Results

The current section numerically validates the proposed SC model. Accordingly, this study considers the numerical data from several well-established articles, like Taleizadeh et al. [9], Saithong and Luong [27], and Daryanto et al. [49]. Here, Table 3 enlists the values (data are scaled in the 1000 s) of various parameters of the proposed model.

**Table 3.** Various parameters of the proposed inventory model.

| Parameters | Values | Parameters | Values |
|:---:|:---:|:---:|:---:|
| $D_i$ | 10 units per unit time | $\beta$ | 0.5 |
| $c_h$ | USD 1 per unit | $c_b$ | USD 30 per unit item |
| $c_p$ | USD 30 per unit item | $c_l$ | USD 40 per unit item |
| $\alpha$ | 0.1 per unit time | $p_i$ | USD 100 per item |
| $p$ | 0.15 | $p_s$ | USD 40 per unit item |
| $T$ | 0.47 day | $c_o$ | USD 100 per order |
| $\epsilon_i$ | 0 | | |

*Optimal Results*

This study computes the optimal base-stock level that maximizes the retailer's expected long-run net profit in any cycle, obtained as follows (* denotes optimality): (the calculation is in Appendix F) (Figure 4)

$$S^* = 5.53 \text{ units and } E(NP^*(S^*)) = \text{USD } 1184.74. \tag{43}$$

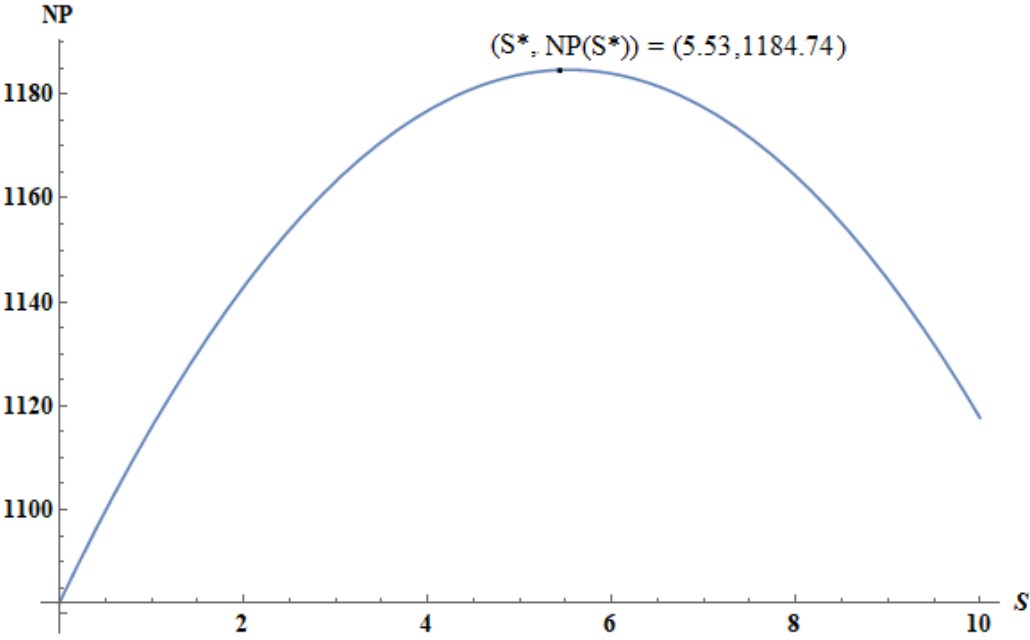

**Figure 4.** Concavity of the long-run net profit with respect to base-stock level.

This optimal result indicates that the proposed SC model is economically viable for the retailer in the long run. Here, this study employs the popular *WOLFRAM MATHEMATICA* (2019) software in one portable computer with an *Intel® CORE^TM i7* processor, 16 GB in RAM, and the Windows 10® operating system.

## 7. Sensitivity Analysis, Managerial Insights, and Comparison of Results

This section performs the sensitivity analysis of several major parameters of the proposed SC model, thereby extracting a number of useful managerial insights. Then, this study compares the current research with some well-established articles in this regard.

### 7.1. Sensitivity Analysis

The optimal expected long-run net profit of the proposed inventory model changes due to various modifications of system parameters, such as demand rate $(D_i)$, selling price

$(p_i)$, cycle length $(T)$, probability of disruption at the supplier side $(p)$, deterioration rate $(\alpha)$, back-order portion $(\beta)$, acquisition cost $(c_p)$, lost sales cost $(c_l)$, and unit holding cost $(c_h)$, with each parameter being changed $-50\%$, $-25\%$, $25\%$, or $50\%$ each time while the other parameters are at their original levels. In this way, this study derives the following observations (see, for results, Table 4 with Figure 5):

- The optimal values of the base-stock level $(S^*)$ in the proposed inventory model exhibit swift escalation (reduction) in response to increases (decreases) in the values of $D$, $T$, and $p$, as well as decreases (increases) in the values of $\beta$ and $\alpha$.
- Optimal expected long-run net profit values for retailers experience a substantial rise (fall) in response to heightened (reduced) values of $D$ and $p_i$.
- Reduced values of parameter $c_p$ correspond to lofty optimal base-stock levels for retailers.
- A higher value of parameter $T$ corresponds to an increased expected long-run net profit for retailers, while an exceedingly low value of $T$ rapidly reduces the expected long-run net profit.
- Retailers' expected long-run net profit changes inversely to the changes in $c_p$.
- The values of $S^*$ are moderately sensitive to any changes in the values of $p_i$ and $c_l$.
- Changes in the parameter value of $c_h$ have a minimal impact on the sensitivity of the values of $S^*$.
- The value of $E(NP)^*$ remains relatively unaffected using variations in $c_h$, $c_l$, $\alpha$, and $p$, while changes in $\beta$ exert minimal influence on the values of $E(NP)^*$.

**Table 4.** Sensitivity analysis of major parameters of the proposed inventory model.

| Parameters (Initial Values) | % Changes | % Changes in $S^*$ | % Changes in $NP^*(S^*)$ |
|---|---|---|---|
| $D_i$ (10) | 50 | 49.99 | 67.96 |
| | 25 | 25 | 33.98 |
| | $-25$ | $-25$ | $-33.98$ |
| | $-50$ | $-50$ | $-67.96$ |
| $p_i$ (100) | 50 | $-7.61$ | 85.52 |
| | 25 | $-3.84$ | 42.75 |
| | $-25$ | 3.91 | $-42.72$ |
| | $-50$ | 7.89 | $-85.41$ |
| $T$ (0.47) | 50 | 23.98 | 6.19 |
| | 25 | 12.14 | 4.22 |
| | $-25$ | $-12.42$ | $-8.71$ |
| | $-50$ | $-25.11$ | $-28.64$ |
| $p$ (0.15) | 50 | 26.82 | 1.73 |
| | 25 | 14.07 | 0.8 |
| | $-25$ | -15.63 | -0.56 |
| | $-50$ | $-33.13$ | $-0.79$ |
| $\alpha$ (0.1) | 50 | $-23.3$ | $-1.91$ |
| | 25 | $-11.42$ | $-1.07$ |
| | $-25$ | 10.98 | 1.29 |
| | $-50$ | 21.55 | 2.78 |

**Table 4.** *Cont.*

| Parameters (Initial Values) | % Changes | % Changes in $S^*$ | % Changes in $NP^*(S^*)$ |
|---|---|---|---|
| $\beta$ (0.5) | 50 | $-19.48$ | 3.91 |
| | 25 | $-10$ | 1.87 |
| | $-25$ | 10.56 | $-1.68$ |
| | $-50$ | 21.75 | $-3.17$ |
| $c_p$ (30) | 50 | $-7.22$ | $-16.65$ |
| | 25 | $-4.28$ | $-8.34$ |
| | $-25$ | 6.84 | 8.39 |
| | $-50$ | 19.49 | 16.89 |
| $c_l$ (40) | 50 | 8.2 | $-1.24$ |
| | 25 | 4.1 | $-0.63$ |
| | $-25$ | $-4.1$ | 0.67 |
| | $-50$ | $-8.2$ | 1.36 |
| $c_h$ (1) | 50 | $-2.51$ | $-0.24$ |
| | 25 | $-1.26$ | $-0.12$ |
| | $-25$ | 1.26 | 0.13 |
| | $-50$ | 2.52 | 0.26 |

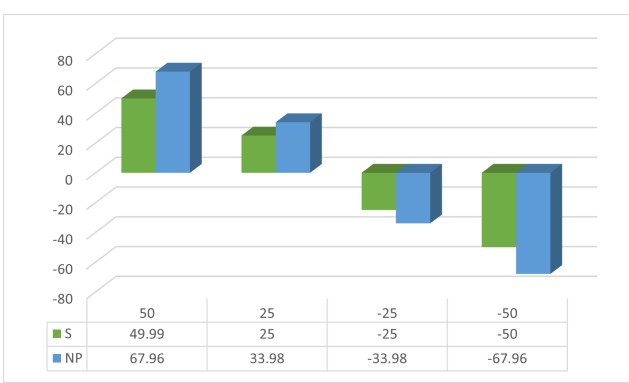

(**a**) Sensitivity analysis of $D_i$

(**b**) Sensitivity analysis of $p_i$

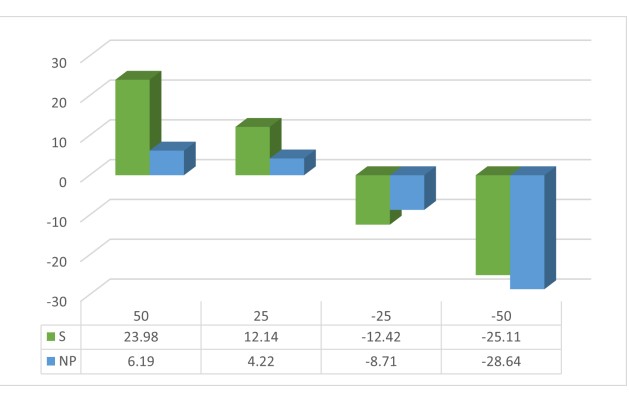

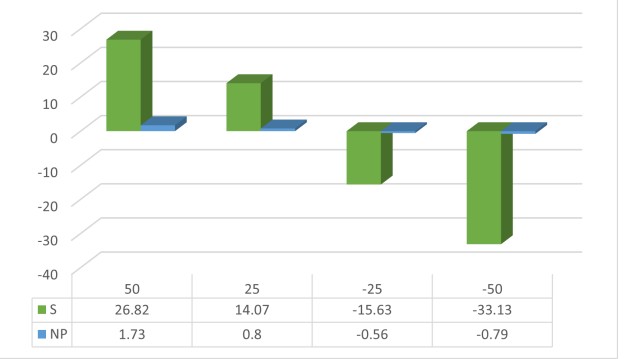

(**c**) Sensitivity analysis of $T$

(**d**) Sensitivity analysis of $p$

**Figure 5.** *Cont.*

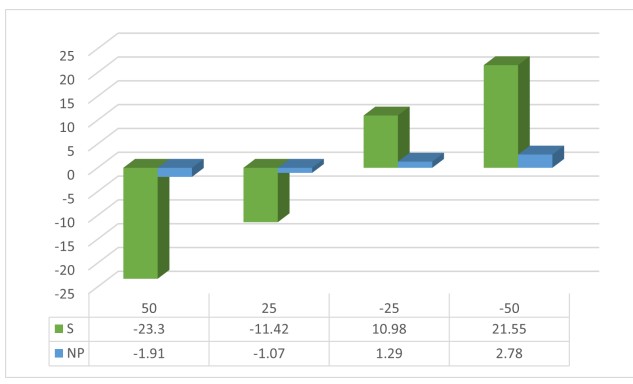

(**e**) Sensitivity analysis of $\alpha$

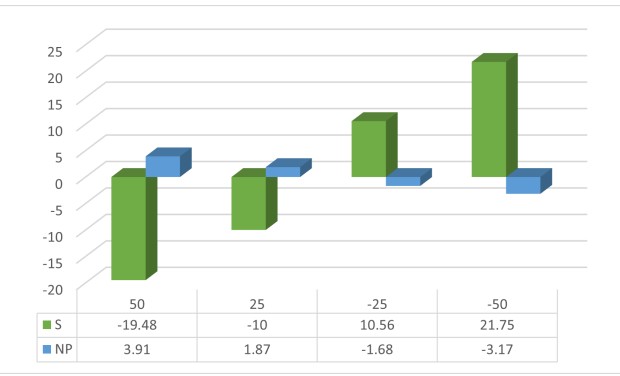

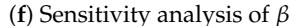

(**f**) Sensitivity analysis of $\beta$

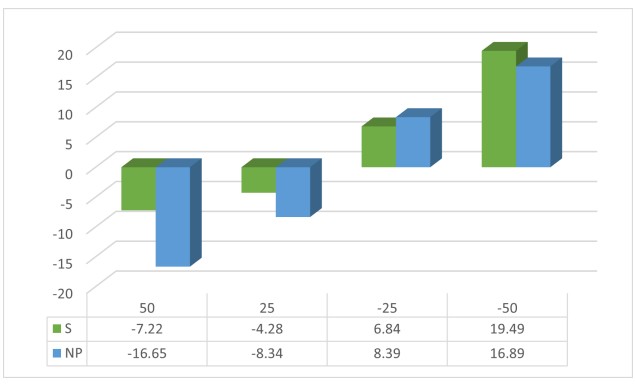

(**g**) Sensitivity analysis of $c_p$

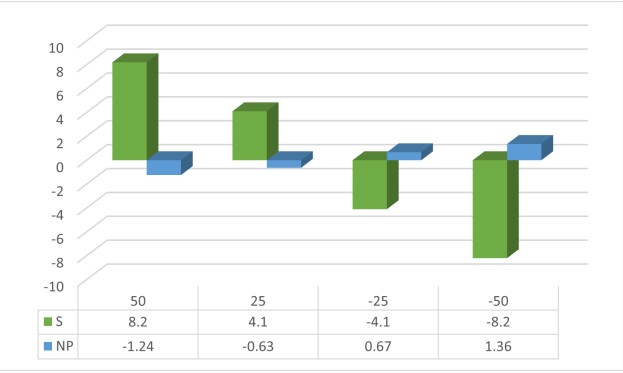

(**h**) Sensitivity analysis of $c_l$

**Figure 5.** Sensitivity analysis of the major parameters of the proposed model.

### 7.1.1. Analysis of Simultaneous Change in $(\alpha, p)$

In this analysis involving the simultaneous alteration of two key parameters, namely the deterioration rate ($\alpha$) set at 0.1 and the probability of supply-side disruption ($p$) set at 0.15, several noteworthy insights have emerged (see Table 5).

Upon examination, it becomes evident that the optimal base-stock level is significantly influenced by the joint manipulation of these parameters. However, intriguingly, the long-run optimal net profit does not exhibit a pronounced and drastic transformation under these conditions.

When confronted with a scenario characterized by both a significant increase in deterioration rate and a heightened probability of supply-side disruptions, it becomes imperative for managers to elevate their base-stock levels. In such circumstances, the consideration of implementing preservation technology becomes particularly relevant, especially if adequate financial resources are available to support this strategic decision.

In cases where a minor alteration in the deterioration rate coincides with a heightened (diminished) likelihood of disruptions, it is consistently advisable to adjust the base-stock level upwards (downwards) to optimize profit potential.

When juxtaposing a higher deterioration rate alongside a lower chance of disruption, these findings indicate that the inventory manager is compelled to maintain a substantially diminished base-stock level. This relationship underscores the crucial role of balancing inventory in response to varying operational dynamics.

Interestingly, when confronted with a scenario where the deterioration rate is exceptionally low while the disruption probability remains high, the inventory manager is compelled to maintain a heightened base-stock level, enabling them to capitalize on the enhanced profit potential despite the potential for disruptions.

**Table 5.** Sensitivity analysis of simultaneous change in $(\alpha, p)$ in the proposed inventory model.

| % Changes of $\alpha$ (Changed Values) | % Changes of $p$ | % Changes in $S^*$ | % Changes in $NP^*(S^*)$ |
|---|---|---|---|
| 50 (0.15) | 50 | 5.96 | −1.23 |
| | 25 | −7.91 | −1.68 |
| | −25 | −40.47 | −1.82 |
| | −50 | −59.8 | −1.28 |
| 25 (0.125) | 50 | 16.58 | 0.15 |
| | 25 | 3.29 | −0.55 |
| | −25 | −27.8 | −1.32 |
| | −50 | −46.17 | −1.17 |
| −25 (0.075) | 50 | 36.7 | 3.48 |
| | 25 | 24.46 | 2.33 |
| | −25 | −3.96 | 0.43 |
| | −50 | −20.64 | −0.14 |
| −50 (0.05) | 50 | 46.25 | 5.42 |
| | 25 | 34.48 | 4.06 |
| | −25 | 7.24 | 1.64 |
| | −50 | −8.68 | 0.76 |

### 7.1.2. Analysis of Simultaneous Change of $(\beta, p)$

In this analysis concerning the simultaneous manipulation of two significant parameters, specifically the back-order portion ($\beta$) set at 0.5 and the probability of supply-side disruption ($p$) set at 0.15, a series of intriguing observations come to light (see Table 6).

**Table 6.** Sensitivity analysis of simultaneous change in $(\beta, p)$ in the proposed inventory model.

| % Changes of $\beta$ (Changed Values) | % Changes of $p$ | % Changes in $S^*$ | % Changes in $NP^*(S^*)$ |
|---|---|---|---|
| 50 (0.75) | 50 | −0.24 | 6.46 |
| | 25 | −9.38 | 5.18 |
| | −25 | −30.76 | 2.71 |
| | −50 | −43.54 | 1.67 |
| 25(0.625) | 50 | 12.82 | 3.93 |
| | 25 | 1.98 | 2.86 |
| | −25 | −23.36 | 1.02 |
| | −50 | −38.42 | 0.42 |
| −25 (0.375) | 50 | 41.85 | −0.09 |
| | 25 | 26.96 | −0.99 |
| | −25 | −7.55 | −2.05 |
| | −50 | −27.67 | −1.94 |
| −50 (0.25) | 50 | 58.05 | −1.48 |
| | 25 | 40.73 | −2.47 |
| | −25 | 0.9 | −3.42 |
| | −50 | −22.03 | −3.05 |

Evidently, the optimal base-stock level is notably responsive to this dual parameter adjustment, underscoring the inherent sensitivity of inventory-management decisions to such variations. However, it is noteworthy that despite the significant influence on the optimal base-stock level, the long-run optimal net profit remains relatively stable, lacking a pronounced and dramatic alteration under these conditions. In scenarios where a higher potential for back orders exists alongside an elevated probability of supply-side disruptions, inventory managers can adhere to their regular base-stock levels to enhance their long-term optimal net profit. Increasing item storage is unnecessary, even in the face of raised disruption probabilities.

Even in scenarios where a higher back-order portion is placed with a very low probability of supply-side disruption, the inventory manager is inclined to maintain an exceptionally reduced base-stock level. This phenomenon emphasizes the trade-off between inventory holding costs and the cost of unmet demand, highlighting the strategic balancing act that inventory managers must navigate.

The analysis demonstrates that when facing a situation characterized by a low back-order portion and a high probability of supply-side disruptions, a substantial augmentation of the base-stock level is imperative to effectively optimize long-term net profit.

### 7.2. Managerial Insights

The aforementioned explorations in Section 7.1 consequently offer a number of insights for average inventory managers. These can be useful in choosing a proficient trade scenario and suppleness scheme throughout the existence of supply-side disruptions and deterioration of products, which are noted as follows.

- As the cement industry maintains a distinct regional focus, the ability of organizations to uphold price discipline amid fierce competition plays a pivotal role in shaping the dynamics of cement bag supply and demand. Recent sensitivity analysis underscores a critical relationship: even the slightest fluctuation in the procurement cost of cement bags wields an inverse impact on the retailer's optimal base-stock level and the resultant anticipated long-term net profit.

  Consequently, retailers are advised to meticulously scrutinize and negotiate before committing to substantial stock purchases from neighboring supplier hubs. On the front of sales strategy, a clear trend emerges: higher price points for cement bags directly correlate with heightened retailer profitability. This revelation prompts various strategic recommendations to empower retailers to enhance their market position. This study advocates for the bundling of products and services, the strategic refinement of the product mix, the phased discontinuation of lower-priced offerings, the meticulous curation of complementary products for specific purchase contexts, targeted employee training initiatives, and the exploration of multifaceted strategies. Moreover, for emerging retailers, an astute approach involves setting a relatively higher sale price, underpinned by a concerted effort to amplify the perceived value of their offerings.

- A substantial rise in the demand rate for cement bags translates to a swift escalation in the retailer's projected long-term net profit. Consequently, the retailer is advised to proactively undertake steps in this trajectory, including enlisting adept technical marketeers, implementing effective market-mapping strategies, tackling customer-centric challenges head-on, and exploring additional measures to capitalize on this potential growth opportunity.

  Furthermore, an uptick in demand for cement bags mandates a notably expanded inventory space requirement for the retailer. As a result, this study advocates for a strategic approach to planned capacity augmentation aimed at counteracting any potential surge in demand.

- In instances where the probability of supply-side disruptions is higher (or lower), this research proposes that retailers should consider bolstering (or trimming) their inventory of cement bags. The retailer's optimal base-stock level also experiences an upward surge during extended review intervals.

Although these scenarios lead to a commensurate rise in the cement retailers' optimal expected long-run net profit, an excessively extended review interval contributes to the further degradation of the cement stored within the bags. Consequently, these compromised bags may go unsold, exacerbated by a subsequent decline in their salvage value.

- In contrast to several existing articles, alterations to either the unit holding cost or the unit lost sales cost exhibit minimal impact on the long-term profitability of cement retailers.
- However, the retailer's optimal base-stock level demonstrates a proportional variation in response to unit shifts in lost sales costs. Therefore, this research recommends that retailers facing constraints in enhancing stock capacity should diligently monitor instances of lost sales, enhance demand forecasting through AI-driven Seasonal Autoregressive Integrated Moving Average (SADS) models, establish contingency suppliers, consider strategic capital investments, and implement other suitable measures.

### 7.3. Comparison of Results

In operations management, effective inventory control ensures optimal resource allocation and cost-efficiency. In this comparison, this study examines two well-established articles with this model that address different aspects of inventory management. The articles Taleizadeh et al. [50] and Saithong and Lekhavat [31] focus on constant demand scenarios and neglect the effect of deterioration. Instead, this work considers selling price-dependent demand and constant deterioration, which is more realistic and essential for accurate modeling in certain industries. Like the above-mentioned articles, this model also considered supply-side disruption, along with partial back-ordering of shortages. The inventory-replenishment policies of Taleizadeh et al. [50] and Saithong and Lekhavat [31] are designed to maintain a fixed stock level to meet efficiently customer demand. While the constant-demand assumption simplifies calculations, it is not an accurate representation of real-world scenarios. Many products experience fluctuating demand due to various factors such as seasonality, market trends, and economic changes. This paper introduces a more realistic inventory-management model that considers both price-dependent demand and deterioration. The variations in selling price can influence demand and, consequently, inventory-replenishment decisions.

In a comparison of managerial insights in the articles of Taleizadeh et al. [50] and Saithong and Lekhavat [31], increased odds of disruptions, such as supply chain bottlenecks or unexpected market shifts, can negatively affect profitability and demand a higher stock level, While in this work as well, a higher rate of disruption reduces the profitability but significantly reduces the base-stock level. In the articles Taleizadeh et al. [50] and Saithong and Lekhavat [31], higher holding costs, such as warehousing expenses, directly impact the optimal base-stock level; on the other hand, in this work, optimal base-stock level and the profits of retailers are not affected for any increases in holding cost. An increase in demand can moderately affect retailers' optimal base-stock level and the profitability of the model Taleizadeh et al. [50], while Saithong and Lekhavat [31] did not investigate this. In this proposed model, any increase in demand rapidly increases the optimal base-stock level and profitability of retailers.

Here, Table 7 compares the current research with some existing and well-established articles by considering both the analytical and the managerial aspects of the proposed inventory model.

**Table 7.** Comparison of current research with well-established articles.

| Aspects | Taleizadeh et al. [9] | Saithong and Lekhavat [31] | This Study |
|---------|----------------------|----------------------------|------------|
| Modeling | Deterministic demand for non-deteriorating items, partial back-ordering of shortages under geometric distribution-based stochastic supply-side disruptions | Deterministic demand for non-deteriorating items, partial back-ordering of shortages under exponential distribution-based stochastic supply-side disruptions | Price sensitive demand for deteriorating cement bags, partial back-ordering of shortages under geometric distribution based stochastic supply-side disruptions |

**Table 7.** *Cont.*

| Aspects | Taleizadeh et al. [9] | Saithong and Lekhavat [31] | This Study |
|---|---|---|---|
| Insights | Increased odds of disruptions reduce the profitability while asking to stock more items, higher holding cost reduces the optimal base-stock level, and soaring demand modestly improves optimal base-stock and profit of retailers | Increased odds of disruptions reduce the profitability (i.e., higher daily cost) while asking to stock more items, higher holding cost reduces the optimal base-stock level, not investigating the impact of demand | Increased odds of disruptions reduce the profitability while significantly reducing the optimal base-stock level, optimal base-stock and profit of retailers are unaffected by increased holding cost, and soaring demand swiftly improves the optimal base-stock and profitability of retailers |

*7.4. Limitations of the Proposed Approach*

- In the classical optimization approach to the nonlinear programming problem, the presence of a large number of decision variables can significantly increase the difficulty in solving them and can result in an impractical amount of time to find a solution.
- Many real-world commerce problems involve complex and nonlinear relationships, multiple objectives, and constraints that may not be easily represented within the framework of classical optimization. Some commerce problems involve discrete decision variables, such as selecting the best combination of products to stock or determining the optimal product mix in a manufacturing process. Classical optimization methods that rely on continuous variables may not handle such discrete decisions efficiently.
- Classical optimization approaches may not fully capture the complexities of human behavior and preferences. In rapidly changing markets, the assumptions made by classical optimization models may quickly become outdated, rendering their solutions less relevant.
- In the case of higher-degree variables, the objective function leads to non-convex optimization problems. In such cases, the function may have multiple local optima, making it difficult to ensure that the solution found is the globally optimal one. Due to higher-degree variables, the search space can grow exponentially, making the search for global optimality computationally expensive.

**8. Conclusions**

The findings of this study have unveiled sustainable base-stock inventory strategies that can effectively enhance the anticipated long-term profitability of cement retailers. These strategies account for the inherent variability in demand, which fluctuates randomly over time, as well as the occasional occurrence of partially back-ordered shortages during stochastic disruptions on the supply side. The proposed decision-support framework has rigorously established the global optimality of the retailer's expected long-run net profit at a strategically determined base-stock level. This achievement is accomplished by factoring in a comprehensive range of cost components and a fixed cycle length. Furthermore, this research underscores the importance of meticulous evaluation and negotiation prior to undertaking substantial stock procurements from neighboring suppliers. It highlights the significance of setting a competitive sales price while enhancing the attractiveness of offerings, along with a proactive approach to planned capacity expansion aimed at mitigating potential increases in demand.

The current research area needs more exploration, as there are some limited assumptions considered. For example, they can determine the most appropriate locations for cement retailers in a staggering market. Also, this work should consider the following points.

- Inventory-management systems should be improved by incorporating constant lead time before disruptions and developing strategies to handle unforeseen events more efficiently. During disruptions, such as natural disasters, supplier issues, transportation problems, or unexpected events, the lead time can become unpredictable and variable. This variability can lead to challenges in inventory management [51].
- Rather than a fixed deterioration rate, the possibility of time-dependent deterioration indicates situations where the quality or usability of inventory items degrades over

time. This can have a significant impact on inventory management and can affect various aspects of the inventory model [52].
- Solutions can be obtained using metaheuristic algorithms rather than classical algorithms. Metaheuristic algorithms are optimization techniques that are particularly useful for solving complex, nonlinear, and combinatorial problems, such as inventory optimization [53].
- The uncertainty aspects can be studied under the environment of Intuitionistic Fuzzy Sets (IFSs) and Neutrosophic Sets (NSs). IFSs and NSs are extensions of classical fuzzy sets and consider the uncertainty and vagueness inherent in real-world inventory-management problems (Wang et al. [54], Barman et al. [55]).

**Author Contributions:** Conceptualization, A.G.; Methodology, A.G., S.K.M. and M.D.; Software, M.D. and M.B.H.; Validation, A.G., C.B.I., S.K.M. and M.B.H.; Writing—original draft, M.D. and A.G.; Writing—review & editing, A.G., C.B.I., and S.K.M.; Visualization, S.K.M. and C.B.I.; supervision, A.G., S.K.M., and C.B.I.; Data curation, M.D. and M.B.H.; Funding acquisition, M.B.H. and C.B.I. All authors have read and agreed to the published version of the manuscript.

**Funding:** This research received no external funding.

**Data Availability Statement:** Not applicable.

**Acknowledgments:** Rimi Karmakar, Dept. of Mathematics, IIEST, Shibpur, India had a pivotal role in the writing and editing purposes during revising this manuscript. Her contributions lead to a considerable improvement of this article and are deeply appreciated.

**Conflicts of Interest:** The authors declare no conflict of interest.

**Appendix A**

This study determines the area under the curve as follows: Area of the $i^{th}$ triangle $\triangle_i$ when the demand rate is $D_i$:

$$\triangle_i = \frac{1}{2}\left(D_i T^2\right). \tag{A1}$$

Area of the $i^{th}$ rectangle $R_i$ when the demand rate is $D_i$:

$$R_i = (S - D_i T)T. \tag{A2}$$

Thus, the area of the $i^{th}$ trapezoid $V_i$ is as follows:

$$A_i = \frac{1}{2}\left(D_i T^2\right) + (S - D_i T)T. \tag{A3}$$

Thus, the total area $\sum_{i=1}^{m-1} A_i$ up to the $(m-1)^{th}$ inventory epoch under the inventory curve is as follows:

$$\sum_{i=1}^{m-1} A_i = \sum_{i=1}^{m-1} \left( \frac{1}{2}\left(D_i T^2\right) + (S - D_i T)T \right). \tag{A4}$$

**Appendix B**

The area of the $(m+r)^{th}$ inventory cycle is as follows:

$$B_{(m+r)} = \frac{\beta D_{m+r}(T - w_2)^2}{2}. \tag{A5}$$

The area of the $(m+r+1)^{th}$ inventory cycle is as follows:

$$B_{(m+r+1)} = \frac{\beta D_{m+r}(T - w_2) + (\beta D_{m+r}(T - w_2) + \beta T D_{m+r+1})T}{2}$$
$$= \left( \beta D_{m+r}(T - w_2) + \frac{\beta T D_{m+r+1}}{2} \right) T. \tag{A6}$$

The area of the $(m+r+2)^{th}$ inventory cycle is as follows:

$$B_{(m+r+2)} = \left(\beta D_{m+r}(T-w_2) + \beta T D_{m+r+1} + \frac{\beta T D_{m+r+2}}{2}\right)T. \tag{A7}$$

The area of the $\tau^{th}, \tau \geq (m+r+2)$ inventory cycle of the proposed model is as follows:

$$B_\tau = \begin{cases} \dfrac{\beta D_{m+r}(T-w_2)^2}{2}, & \text{if } \tau = m+r, \\[2mm] \left(\beta D_{m+r}(T-w_2) + \dfrac{\beta T D_{m+r+1}}{2}\right)T, & \text{if } \tau = m+r+1, \\[2mm] \left(\beta D_{m+r}(T-w_2) + \displaystyle\sum_{i=m+r+1}^{\tau-1} \beta T D_i + \dfrac{\beta T D_\tau}{2}\right)T, & \text{if } \tau \geq (m+r+2). \end{cases} \tag{A8}$$

Thus, this study computes the expected average back-order of the retailer as follows:

$$E(B_\tau) = \sum_{\tau=m+r}^{\infty} \pi_\tau B_\tau = \pi_{m+r}B_{m+r} + \pi_{m+r+1}B_{m+r+1} + \sum_{\tau=m+r+2}^{\infty} \pi_\tau B_\tau. \tag{A9}$$

**Appendix C**

The current study finds the following:

$$f_1(S) = (V_m + V_{m+1} + \dots + V_{m+r-1})$$
$$= (V_{m+r-1} + V_{m+r-2} + \dots + V_m)$$
$$= (1-\alpha T)(V_{m+r-2} + V_{m+r-3} + \dots + V_m) + ST - T^2 \sum_{i=m}^{m+r-2} D_i - \frac{T^2}{2} D_{m+r-1}$$
$$= (1-\alpha T)^{r-1}V_m + ST(1 + (1-\alpha T) + \dots + (1-\alpha T)^{(r-2)}) - \frac{T}{\alpha}\sum_{i=0}^{r-2} D_{m+i}(1-(1-\alpha T)^{(r-i-1)}) - \tag{A10}$$
$$\frac{T^2}{2}\left(D_{m+r-1} + (1-\alpha T)D_{m+r-2} + (1-\alpha T)^2 D_{m+r-3} + \dots + (1-\alpha T)^{r-2}D_{m+1}\right)$$
$$= (1-\alpha T)^{r-1}\left(ST - \frac{D_m T^2}{2}\right) + \frac{S}{\alpha}(1-(1-\alpha T)^{r-1}) - \frac{T}{\alpha}\sum_{i=0}^{r-2} D_{m+i}(1-(1-\alpha T)^{r-1-i}) -$$
$$\frac{T^2}{2}\left(\sum_{i=1}^{r-1} D_{m+r-i}(1-\alpha T)^{i-1}\right).$$

**Appendix D**

In this study, we deduce the following:

$$\sum_{\tau=0}^{r-1} V_{m+\tau}\pi_\tau = V_m\pi_0 + V_{m+1}\pi_1 + \dots + V_{m+r-2}\pi_{r-2} + V_{m+r-1}\pi_{r-1} = f_2(S). \tag{A11}$$

Since $\pi_\tau = p^\tau(1-p), \tau = 0, 1, 2, \dots$ we obtain the following:

$$f_2(S) = (1-p)(p^{r-1}V_{m+r-1} + p^{r-2}V_{m+r-2} + \dots + pV_{m+1} + V_m)$$
$$= (1-p)[p^{r-1}(-\alpha T)(V_{m+r-2} + V_{m+r-3} + \dots + V_m) +$$
$$P^{r-2}(-\alpha T)(V_{m+r-3} + V_{m+r-4} + \dots + V_m) + \dots + p(-\alpha T)V_m + V_m] + STp(1-p^{r-1}) -$$
$$T^2\sum_{i=1}^{r-1} D_{m+i-1}p^i(1-p^{r-i}) - \frac{T^2}{2}(1-p)\sum_{i=1}^{r-1} D_{m+i}p^i$$

Using Appendix C, this study finds the following:

$$
\begin{aligned}
&= (1-p)\Big[(-\alpha T)\Big[p^{r-1}(1-\alpha T)^{r-2}V_m + p^{r-2}(1-\alpha T)^{r-3}V_m + \cdots + pV_m\Big] + V_m\Big] \\
&\quad - ST\Big(\Big[\sum_{i=1}^{r-2} p^{r-i}(1-(1-\alpha T)^{r-1-i})\Big] \\
&\quad + T^2\sum_{j=1}^{r-2}\sum_{i=0}^{r-j-2} D_{m+i}(1-(1-\alpha T)^{r-j-i-1})p^{r-j} + \frac{\alpha T^3}{2}\Big(\sum_{k=2}^{r-1} D_{m+r-k}\sum_{i=1}^{k-1} p^{r-k+i}(1-\alpha T)^{i-1}\Big) \\
&\quad + STp(1-p^{r-1}) - T^2\sum_{i=1}^{r-1} D_{m+i-1}p^i(1-p^{r-i}) - \frac{T^2}{2}(1-p)\sum_{i=1}^{r-1} D_{m+i}p^i \\
&= (1-p)\Big[1 - \frac{p\alpha T}{1-\alpha T}\Big(\frac{1-(p(1-\alpha T))^{r-1}}{1-p(1-\alpha T)}\Big)\Big](ST - \frac{D_m T^2}{2}) \\
&\quad - ST\Big(\Big[\sum_{i=1}^{r-2} p^{r-i}(1-(1-\alpha T)^{r-1-i})\Big] \\
&\quad + T^2\sum_{j=1}^{r-2}\sum_{i=0}^{r-j-2} D_{m+i}(1-(1-\alpha T)^{r-j-i-1})p^{r-j} + \frac{\alpha T^3}{2}\Big(\sum_{k=2}^{r-1} D_{m+r-k}\sum_{i=1}^{k-1} p^{r-k+i}(1-\alpha T)^{i-1}\Big) \\
&\quad + STp(1-p^{r-1}) - T^2\sum_{i=1}^{r-1} D_{m+i-1}p^i(1-p^{r-i}) - \frac{T^2}{2}(1-p)\sum_{i=1}^{r-1} D_{m+i}p^i.
\end{aligned}
\tag{A12}
$$

## Appendix E

$$
\begin{aligned}
&E(NP) = \\
&\frac{1}{T}\Bigg(\Bigg(\sum_{i=1}^{m} p_i D_i T + \sum_{i=1}^{r-1}\pi_i p_{m+i}D_{m+i}T + \pi_r p_{m+r}D_{m+r}w_2 + \pi_r D_{m+r}(T-w_2)p_{m+r}\beta + \\
&\sum_{i=r+1}^{\infty}\pi_i\beta D_{m+i}p_{m+i}T\Bigg) - \Bigg(\Big(c_o(m-1) + c_p\Big(\sum_{i=1}^{m-1}(D_i T + \alpha(\tfrac{1}{2}D_i T^2 + (S-D_i T)T))\Big)\Big) + \\
&c_h\sum_{i=1}^{m-1}\Big(\tfrac{1}{2}\big(D_i T^2\big) + (S-D_i T)T\Big) + \alpha\sum_{i=1}^{m-1}(p_i - p_{s_i})\Big(\tfrac{1}{2}\big(D_i T^2\big) + (S-D_i T)T\Big)\Bigg) + (c_o + \\
&c_p\Big(\sum_{\tau=0}^{r-1}\frac{D_{m+\tau}T^2}{2} + \alpha f_2(S) + \frac{\big(S - \sum_{i=m}^{m+r-1}D_i T - \alpha f_1(S)\big)^2}{2D_{m+r}} + \\
&\frac{1}{2}(1-p)p^r\beta D_{m+r}\Big(T - \frac{\big(S - \sum_{i=m}^{m+r-1}D_i T - \alpha f_1(S)\big)}{D_{m+r}}\Big)^2 + \\
&\frac{1}{2}T^2\sum_{\tau=r+1}^{\infty}\beta D_{m+\tau}\pi_\tau + \beta D_{m+r}T(T - \frac{\big(S - \sum_{i=m}^{m+r-1}D_i T - \alpha f_1(S)\big)}{D_{m+r}})p^{r+1} + \\
&T^2\sum_{\tau=r+2}^{\infty}\pi_\tau\sum_{i=r+1}^{\tau-1}\beta D_{m+i}) + c_h\Bigg(f_2(S) + (1-p)p^r\frac{\big(S - \sum_{i=m}^{m+r-1}D_i T - \alpha f_1(S)\big)^2}{2D_{m+r}}\Bigg) + \\
&\sum_{i=m}^{m+r-1}\Bigg((p_i - p_{s_i})\alpha\Bigg(f_2(S) + (1-p)p^r\frac{\big(S - \sum_{i=m}^{m+r-1}D_i T - \alpha f_1(S)\big)^2}{2D_{m+r}}\Bigg)\Bigg) + \\
&c_b\Big(\tfrac{1}{2}(1-p)p^r\beta D_{m+r}\Big(T - \frac{\big(S - \sum_{i=m}^{m+r-1}D_i T - \alpha f_1(S)\big)}{D_{m+r}}\Big)^2 + \tfrac{1}{2}\beta T^2\sum_{\tau=r+1}^{\infty}D_{m+\tau}\pi_\tau + \\
&p^{r+1}\beta D_{m+r}\Big(T - \frac{\big(S - \sum_{i=m}^{m+r-1}D_i T - \alpha f_1(S)\big)}{D_{m+r}}\Big)T + \beta T^2\sum_{\tau=r+2}^{\infty}\pi_\tau\sum_{i=r+1}^{\tau-1}D_{m+i}) + \\
&c_l\Bigg((1-\beta)p^r D_{m+r}\Big(T - \frac{\big(S - \sum_{i=m}^{m+r-1}D_i T - \alpha f_1(S)\big)}{D_{m+r}}\Big) + \\
&T\sum_{\tau=r+1}^{\infty}\pi_\tau\sum_{i=r+1}^{\tau}(1-\beta)D_{m+i}\Bigg)\Bigg)\Bigg)\Bigg).
\end{aligned}
\tag{A13}
$$

**Appendix F**

By substituting the value of the parameters of Table 3 into Equation (35), we transform the long-run net profit with respect to variable $S$, and it is expressed as

$$E(NP(S)) = -3.35433S^2 + 37.1061S + 1082.12 \qquad (A14)$$

Here, the optimal base-stock level does not depend on the constant term in Equation (A14). So considering, $E(NP_1(S)) = -3.35433S^2 + 37.1061S$ and differentiating equation $E(NP_1(S))$ with respect to $S$, we achieve the first derivative equation.

$$\frac{d(E(NP_1(S)))}{dS} = 37.1061 - 6.7086S \qquad (A15)$$

The necessary and sufficient condition for the critical value is $\frac{d(E(NP_1(S)))}{dS} = 0$. The critical value of the base-stock level is obtained from Equation (A15).

$$S^* = 5.53 \qquad (A16)$$

Differentiating Equation (A15) again with respect to $S$, we find that the long-run net profit is optimal for $S^* = 5.53$.

$$\frac{d^2E(NP(S))}{dS^2} = -6.7086 < 0 \qquad (A17)$$

Also, the optimal net profit is as follows: $NP^*(S^*) =$USD $1,184.74$.

**Appendix G**

$$B_1 = \frac{1}{2(1-p)(1-\alpha T)(c_b+c_p)}(Tc_b(D_2(1-p)(1-\alpha T)+2D_3)-$$
$$2D_3(c_l+(1-p)p_3)+Tc_p(D_2(p-1)(\alpha T-2)+2D_3)). \qquad (A18)$$

$$B_2 = \frac{1}{(\alpha T-1)^2((1-p)pc_h+c_p+\alpha(1-p)p(p_2-p_{s_2}))} \times$$
$$\left(D_3 T\left(\frac{D_2(p-1)pc_h(\alpha T-2)(\alpha T-1)}{2D_3} - \frac{D_2c_p(\alpha^2 T^2-3\alpha T+2)}{2D_3} - (p-1)c_h+ \right.\right. \qquad (A19)$$

$$c_h + \frac{pc_l(\alpha T-1)}{T} - \alpha pc_p + 2\alpha c_p + \frac{\alpha D_2(p-1)p(p_2-p_{s_2})(\alpha T-2)(\alpha T-1)}{2D_3} +$$
$$\left.\left.\alpha(p_1-p_{s_1})-\alpha(p-1)(p_2-p_{s_2})-\frac{(p-1)pp_3(\alpha T-1)}{T}\right)\right). \qquad (A20)$$

$$B_3 = \frac{(p-1)p(c_b+c_p)}{(1-p)pc_h+c_p+\alpha(1-p)p(p_2-p_{s_2})}. $$

$$C_1 = \frac{1}{2p(\alpha T-1)^2}(D_2 pT(\alpha T-2)(\alpha T-1)-2D_3 T+$$
$$\frac{1}{(p-1)p(\beta(c_b+c_p)+c_h+\alpha(p_2-p_{s_2}))-c_p}\left(2D_3\left(\beta pTc_b(\alpha pT-1)+pTc_h+\alpha\beta p^2T^2c_p- \right.\right. \qquad (A21)$$
$$\alpha p^2 Tc_p+2\alpha pTc_p-\beta pTc_p-Tc_p-\beta p_3 p^3+\alpha\beta p_3 p^3 T+\alpha p_3 p^3(-T)+p_3 p^3+\beta p_3 p^2-$$
$$\left.\left.\alpha\beta p_3 p^2 T+\alpha p_3 p^2 T-p_3 p^2-\alpha pTp_{s_1}+\alpha p_1 pT\right)\right)).$$

$$C_2 = \frac{(1-\beta)D_3 p}{(\alpha T-1)((p-1)(\beta pc_b+pc_h+\beta pc_p+\alpha p(p_2-p_{s_2}))-c_p)} \qquad (A22)$$

$$E_1 = \frac{T\left(D_2\left(\beta p^2 - \beta p - 1\right)\left(\alpha^2 T^2 - 3\alpha T + 2\right) + 2D_3(\alpha(p(\beta T - 1) + 2) - \beta p)\right)}{2(\beta p^2 - \beta p - 1)(\alpha T - 1)^2}. \quad \text{(A23)}$$

$$E_2 = \frac{1}{2D_3(1 + \beta p - \beta p^2)}\Big((p - 1)p\left(D_2\left(\beta p^2 - \beta p - 1\right)\left(\alpha^2 T^2 - 3\alpha T + 2\right) + \\
2D_3(\alpha(p(\beta T - 1) + 2) - \beta p))(\beta c_b + c_h + \alpha(p_2 - p_{s_2}))\right) + \frac{\beta D_2(1 - p)pc_b(\alpha T - 1)}{D_3} + \\
\frac{\alpha\beta D_2(p - 1)pTc_b(\alpha T - 1)}{2D_3} - \beta p^2 c_b(1 - \alpha T) + \beta(1 - p)pc_b(\alpha T - 1) + \\
\frac{D_2(1 - p)pc_h(\alpha T - 1)}{D_3} - \frac{\alpha D_2(1 - p)pTc_h(\alpha T - 1)}{2D_3} + (1 - p)c_h + c_h + \\
\frac{(1 - \beta)pc_l(\alpha T - 1)}{T} + \frac{\alpha^2 D_2(p - 1)pT(p_2 - p_{s_2})(\alpha T - 1)}{2D_3} + \\
\frac{\alpha D_2(1 - p)p(p_2 - p_{s_2})(\alpha T - 1)}{D_3} + \alpha(p_1 - p_{s_1}) + \alpha(1 - p)(p_2 - p_{s_2}) + \\
\frac{\beta(p - 1)p_3 p(\alpha T - 1)}{T} + \frac{(1 - p)p_3 p(\alpha T - 1)}{T}. \quad \text{(A24)}$$

$$E_3 = \frac{\alpha\beta(1 - p)pc_b(1 - \alpha T)}{D_3} + \frac{\beta(p - 1)pc_b(1 - \alpha T)}{D_3 T} + \frac{\alpha(1 - p)pc_h(1 - \alpha T)}{D_3} + \\
\frac{(p - 1)pc_h(1 - \alpha T)}{D_3 T} + \frac{\alpha^2(1 - p)p(p_2 - p_{s_2})(1 - \alpha T)}{D_3} + \frac{\alpha(p - 1)p(p_2 - p_{s_2})(1 - \alpha T)}{D_3 T}. \quad \text{(A25)}$$

$$E_4 = \frac{\alpha\beta(1 - p)p(1 - \alpha T)}{D_3} + \frac{\beta(p - 1)p(1 - \alpha T)}{D_3 T} + \frac{\alpha(1 - \alpha T)}{D_3} - \frac{1 - \alpha T}{D_3 T}. \quad \text{(A26)}$$

$$F_1 = \frac{T\left(D_2(p - 1)p\left(\alpha^2 T^2 - 3\alpha T + 2\right) - 2D_3(p - 2)\right)}{2(p - 1)p(\alpha T - 1)^2}. \quad \text{(A27)}$$

$$F_2 = \frac{1}{(1 - p)pT}\Big(-\beta(p - 1)pTc_b(\alpha pT - 2) + (\beta - 1)(p - 1)p^2 c_l(\alpha T - 1) - \alpha\beta p^3 T^2 c_p + \\
\alpha p^3 Tc_p + \alpha\beta p^2 T^2 c_p - 3\alpha p^2 Tc_p + 2\beta p^2 Tc_p + 2\alpha pTc_p - 2\beta pTc_p + pTc_p - 2Tc_p + \beta p_3 p^4 - \\
\alpha\beta p_3 p^4 T + \alpha p_3 p^4 T - p_3 p^4 - 2\beta p_3 p^3 + 2\alpha\beta p_3 p^3 T - 2\alpha p_3 p^3 T + 2p_3 p^3 + \beta p_3 p^2 + \\
\alpha p^2 Tp_{s_1} - \alpha p^2 Tp_{s_2} - \alpha\beta p_3 p^2 T - \alpha p_1 p^2 T + \alpha p_2 p^2 T + \alpha p_3 p^2 T - p_3 p^2 - \alpha pTp_{s_1} + \\
\alpha pTp_{s_2} + \alpha p_1 pT - \alpha p_2 pT\Big). \quad \text{(A28)}$$

$$F_3 = \frac{\alpha\beta(1 - p)pc_b(1 - \alpha T)}{D_3} + \frac{\beta(p - 1)pc_b(1 - \alpha T)}{D_3 T} + \frac{\alpha\beta(1 - p)pc_p(1 - \alpha T)}{D_3} + \\
\frac{\beta(p - 1)pc_p(1 - \alpha T)}{D_3 T} + \frac{\alpha c_p(1 - \alpha T)}{D_3} - \frac{c_p(1 - \alpha T)}{D_3 T} + \frac{\alpha^2(1 - p)p(p_2 - p_{s_2})(1 - \alpha T)}{D_3} + \\
\frac{\alpha(p - 1)p(p_2 - p_{s_2})(1 - \alpha T)}{D_3 T}. \quad \text{(A29)}$$

$$F_4 = \frac{\alpha(1 - p)p(1 - \alpha T)}{D_3} + \frac{(p - 1)p(1 - \alpha T)}{D_3 T}. \quad \text{(A30)}$$

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
