# Peer review of "Optimal Base-Stock Inventory-Management Policies of Cement Retailers under Supply-Side Disruptions"

_mathematics, doi:10.3390/math11183971_

Round 1
Reviewer 1 Report
The authors proposed a model to compute an industry's optimal base stock level.
Some comments/suggestion are as follow:
1. About the practicality of the proposed approach. Eq. 32 is the average long-run profit per cycle. But how many COVID pandemics must happen in the long run to achieve your results? Or what kind of disturbances must take place to achieve your results?
2. Eq. 1, what is $\rho$?
3. You have to solve d[E(NP)]/dS to find the S* given Lemma 1. How do you solve it to find Eq. 33? Do you have insight about the concave shape of your model?
4. From Table 4 and Fig. 3, it seems that only one parameter can change at a time. Is it true? Is it practical?
5. A better explanation of the equations is thankful.
Author Response
.

Reviewer 2 Report
The objective of the this research is to find the most effective inventory management strategies for cement retailers that can lead to the highest possible long-term profits in the event of supply-side disruptions. The suggested economic order quantity model takes into account both intermittent changes in demand for perishable goods and incomplete backorders that may arise due to these unpredictable disruptions. The authors have done well to compile this manuscript. However, there are certain issues needing attention to improve on the paper. The following comments must be considered carefully to revise the paper:
1. The language of the paper should be improved to ensure a better reading.
2. The contributions of the paper should be enhanced.
3. Can the authors specify if the case study is a real-case or an illustrative example.
4. If it is a real-case study, the authors can provide the background information of the experts.
5.The research approach of this study is not clear.
6. There should be some discussions on the limitations of the used methods.
7. Improvement of the conclusions section and definition of avenues for future work.
8. What is the expected long-run shortage cost in the proposed inventory model for the cement industry when the shortages start to happen after the sequential disruptions reach the (m + r)th inventory cycle?
9. How is the expected long-run shortage cost, lost sales cost and backordering cost of the retailer measured and what cost components does it consist of?
10. How are these costs used to compute the expected long-run net profit per cycle and the expected long-run aggregate cost of the retailer?
11. How does the study's modeling and insights compare to those of previous research, as presented in Table 5?
12. How do changes in demand rate, cycle length, probability of disruption at supplier, deterioration rate, backorder rate, acquisition cost, lost sales cost, and unit holding cost affect the optimal base-stock level and expected long-run net profit of cement retailers?
13. How do these observations provide insights for inventory managers in the cement industry?
The objective of the this research is to find the most effective inventory management strategies for cement retailers that can lead to the highest possible long-term profits in the event of supply-side disruptions. The suggested economic order quantity model takes into account both intermittent changes in demand for perishable goods and incomplete backorders that may arise due to these unpredictable disruptions. The authors have done well to compile this manuscript. However, there are certain issues needing attention to improve on the paper. The following comments must be considered carefully to revise the paper:
1. The language of the paper should be improved to ensure a better reading.
2. The contributions of the paper should be enhanced.
3. Can the authors specify if the case study is a real-case or an illustrative example.
4. If it is a real-case study, the authors can provide the background information of the experts.
5.The research approach of this study is not clear.
6. There should be some discussions on the limitations of the used methods.
7. Improvement of the conclusions section and definition of avenues for future work.
Author Response
.

Round 2
Reviewer 1 Report
The work has been improved since the last revision. The feedback has clarified all my concerns.
Reviewer 2 Report
Comments have been addressed carefully and sufficiently, the revisions are rational from my point of view, I think the current version of the paper can be accepted.
Comments have been addressed carefully and sufficiently, the revisions are rational from my point of view, I think the current version of the paper can be accepted.